# Altimeter Observations of Tropical Cyclone-Generated Sea States: Spatial Analysis and Operational Hindcast Evaluation

**Clarence Collins** [1,*] , **Tyler Hesser** [2] , **Peter Rogowski** [3] and **Sophia Merrifield** [3]

1   Coastal and Hydraulics Laboratory, U.S. Army Engineering Research and Development Center, Duck, NC 27949, USA

2   Coastal and Hydraulics Laboratory, U.S. Army Engineering Research and Development Center, Vicksburg, MS 39180, USA; Tyler.Hesser@usace.army.mil

3   Coastal Observing R&D Center, Marine Physical Laboratory, Scripps Institution of Oceanography, La Jolla, CA 92093, USA; progowski@ucsd.edu (P.R.); smerrifield@ucsd.edu (S.M.)

*   Correspondence: Clarence.O.Collins@usace.army.mil

**Abstract:** Tropical cyclones (TC) are some of the most intense weather systems on Earth and are responsible for generating hazardous waves on the sea surface that dominate the extreme wave climate in several regions, including the Gulf of Mexico and the U.S. East Coast. Modeling these waves is crucial for engineering applications, yet it is notoriously difficult, due to TC's compact structure and rapid evolution in space and time relative to other weather systems. To better understand the wave structure under TCs, we use satellite altimeter data paired with TC tracks. We parse the data by TC intensity and forward translation velocity, finding evidence of extended fetch. We use the altimeter data to evaluate operational hindcasts, including the US Army Corps of Engineer's Wave Information Study, National Oceanic and Atmospheric Administration's National Centers for Environmental Prediction Production Hindcast, and the Institut français de recherche pour l'exploitation de la mer (Ifremer) hindcast. The Ifremer hindcast (1990–2016) is examined in detail. Near the eye in the TC-centered reference frame, we find a pattern of model underestimation in the right sector and over estimation in the left and back sectors. This pattern holds, albeit modulated, across various intensities, forward translation velocities, and radii of maximum winds; the exceptions being the most intense and smallest storms, where underestimation is more severe and expands to all sectors near the TC eye.

**Keywords:** tropical cylcone; hurricane; ocean wave; wave dispersion; significant wave height; satellite altimeter; wave model; hindcast; fetch

## 1. Introduction

Tropical cyclones (TC) are compact, intense low-pressure systems (LPSs). TCs and the waves generated by these storms[1] have fascinated scientists and seafarers for generations [1,2]. TCs are critically important to our society [3], as they affect coastal habitation, design of marine structures, oil infrastructure, and shipping. TCs can generate extreme ocean waves. TC-generated waves dominate the wave climate in several areas worldwide, notably the U.S. East Coast and the Gulf of Mexico. The impact of TC-generated waves, which manifests each hurricane season, cannot be overstated: surge, flooding, impact to protective natural features such as dunes, shifting sediments, and damage to infrastructure. Therefore, it is paramount that our operational modeling systems are able to accurately simulate the waves generated by these storms. Forward-looking models are important for planning and preparation and hindcasts are necessary for engineering design considerations. Here, we focus on multi-decade-long operational hindcasts.

---

1   We will use TCs and storms interchangeably throughout this text. Note that TCs are conventionally referred to as hurricanes in the North Atlantic Basin and typhoons in the Western Pacific Basin.

In this study, we compare wave heights measured by satellite altimeters and modeled by operational hindcasts. By combining data from hundreds of storms from over 20 years, we find coherent and consistent error patterns in a storm-coordinate reference frame. Analogous to missing the peak of a time series, we find that the region of highest wave height is systematically underestimated by operational hindcasts. Results are consistent across three hindcasts. These findings support previous studies [4] and our current understanding of wave generation in TCs.

Although modern wave models have shown high skill (e.g., [5,6]), seas generated by TCs offer a special set of challenges. This study was partially motivated by the tendency of wave models to "miss the peak", an issue well known in the wave modeling community (e.g., [7]). Cavaleri (2009) [7] explores the myriad of possible causes. One cause is model physics that are either poorly understood (e.g., shift of physics in extreme environments) or poorly represented (e.g., resolution, advection schemes). The most prevalent explanation is the forcing: deficiencies inherent in the wind field used to drive the waves. The wind fields are a product of modeling mired by analogous issues. In particular, reanalysis winds have been shown to systematically underestimate TC intensity [8,9].

Extreme winds are produced by other types of LPSs. What makes a TC unique is the structure of the wind field. The wind field tends to be very compact around a centralized area of low pressure, and flows inward, like a vortex, while the entire system translates and evolves through space and time. From a TC-centered point of view, the inward flow rotates counter-clockwise (in the Northern Hemisphere), with wind speeds that can be described as a function of radial distance: at the TC center, the wind speed is low, even calm, but ramps up to a maximum in the violent eyewall; from the eyewall, the wind speed tapers down (e.g., [10,11]). Wind speeds are higher on the right-hand side of the storm where forward propagation of the storm is aligned with the wind direction, and vice versa on the left-hand side.

Wave heights under TCs have been found to be asymmetric, due to (1) the wind field asymmetry [12] and (2) asymmetric duration of wind input. Waves on the right side receive extended duration of high winds (where waves on the left side receive shortened duration) [13]. In other words, the joint movement of waves and storm increase the effective fetch-the fetch and duration, being the area and time over which winds are actively inputting energy into waves. This is known as the dynamic fetch phenomenon[2] and it occurs when wave group velocity is similar to the translational speed of the storm. It is interesting to note, though often overlooked, that this phenomenon was described in some of the earliest studies [1,14,15], as discussed in [16]. Subsequently, there has been much progress in verifying and quantifying dynamic fetch and developing parametric wave models that reproduce it (e.g., [12,13,17–22]). In a TC-centered frame of reference, the highest waves are generated on the right side, and tend to emanate out, contributing to the seas elsewhere around a storm, such that, far from the center of a TC wave spectra, are typically bi-modal locally generated wind-seas with remotely generated swells. The swell propagates along ray paths out from the center, while wind-sea is typically aligned with the local wind direction [4,23–25] (as illustrated in Figure 3 of [26]).

The Coupled Boundary Layer Air-Sea Transfer Experiment (CBLAST) was a comprehensive study of waves and air-sea interaction during TCs. Black et al. (2007) [27] present an overview of the observations, which included scanning radar altimeter (SRA) observations [28]. The SRA was mounted on hurricane hunter aircraft and, over the years, has flown through a limited number of storms. After significant processing, SRA provided directional spectra for frequencies up to 0.17 Hz along the flight path. The analysis and subsequent reanalysis (e.g., [4,26]) of SRA data show a pattern of different types of directional spectra that occur with some consistency around a TC. Within a TC-centered frame of reference, Black et al. (2007) [27] proposed 3 azimuthal divisions of space called sectors: relative to the direction of the storm's forward translation at 0°, a left sector (240–20°), a

---

2　Other terms for this can be found in the literature: trapped fetch, extended fetch, fetch resonance, traveling fetch, etc.

right sector (20–150°), and a back sector (50–240°). Depending on the relative angle between the wind and dominant waves, one can define the sea state as wind-sea (peak wave celerity slower than the wind speed and more or less aligned with the wind), following swell (peak wave celerity faster than the wind speed, more or less aligned with the wind), cross-swell (peak wave celerity propagating perpendicular to the wind direction), opposing-swell (peak wave celerity propagating opposed to the wind direction) [29]. The near-field sectors have been found to be dominated by cross-swell (CS) in the left sector, following-swell (FS) in the right sector, and opposing-swell (OS) in the back sector [26–28].

Spectral wave models have been shown to have spatial error patterns that correspond with these sectors. Liu et al. (2017) [4] documented the performance of two models and several source term (ST) physics during Hurricane Ivan (2004). Even for STs that performed relatively well, they found a consistent overestimation of CS in the left sector and OS in the back sector. Turning up a wind-induced swell dissipation parameter improved the performance of one source term (ST6), suggesting the need for better (or better tuned) physics in the models.

The primary aim of the present study is to further explore the shortcomings of operational wave models in TC environments. To do this, we gathered all the altimeter observations of TCs in the North Atlantic, which include data from a number of satellite missions in hundreds of different storms over many years (see Section 2). A companion study considers buoy data (including spectra), for several case studies closer to the coast (Rogowski et al., this issue[3]). With an abundance of altimeter data, we are able to (1) demonstrate how the spatial distribution of wave height around a storm varies with maximum wind speed and forward translation velocity, which builds upon the observations of [22] (Section 3.1), and (2) calculate robust error statistics for three operational hindcasts (Section 3.2). In Section 3.3, we take a deep dive into the Ifremer hindcast (1990–2016). We show that, to an extent, the results of [4] hold up in general. Wave heights are underestimated in the right sector and overestimated elsewhere (Section 3.3.4). This pattern varies with maximum wind speed, forward translation velocity, and radius of maximum winds.

## 2. Methods

To amass the requisite data, we took advantage of publicly available hurricane best track information, curated altimeter databases, and wave model hindcast output, as described below.

### 2.1. TC Information

LPS data have been collected by various agencies around the world. The most well observed basin on Earth is the Atlantic Ocean, and most of these data were collected by the U.S. National Hurricane Center. The databases provide time and coordinates of the center of the LPS, and when possible, information about its structure, i.e., radius of maximum wind, $rmw$, and maximum sustained wind speed, $U_r$. In addition, the storm orientation and forward translation velocity is often provided. When these are not provided, they can be found by calculating the vector between two successive coordinates. The velocity is vector magnitude divided by the time step, and the direction of propagation is the orientation of the vector. The direction of propagation is used to rotate the measurements and the $rmw$ is used to normalize the distance. Through this method, observations are placed in a frame of reference with the TC in the center propagating up the page. This TC-centered reference frame provides the spatial context for model error in terms of the storm sectors previously described.

The preferred source of LPS information was the Tropical Cyclone Observations-Based Structure (TC-OBS) database [30]. We used the version 0.40 of this dataset, which covers the time period of 1995–2014. TC-OBS has enhanced temporal and spatial resolution, often hourly, based on the reanalysis of observations. To cover other potential periods

---

3    Rogowski et al., Performance Assessment of Hurricane Wave Hindcats. *JMSE*. Under Revision.

of interest, 1985–1995 and 2014–2018, we used the International Best Track Archive for Climate Stewardship (IBTrACS) database [31,32]. IBTrACS contains similar information at a 6 h resolution. When used, IBTrACS data were linearly interpolated to an hourly resolution. These two databases track all low pressure systems (LPS), not necessarily of TC strength, which is $U_r \geq 33$ m/s (see Table 1). Altogether, this resulted in over 500 individually identified LPSs, and for 481 of them, there were one or more altimeter passes within 500 km at some point during the LPS life cycle. The internationally recognized S-S scale was designed to indicate potential damage of landfalling TCs and is based on wind speed alone (Table 1). Once a TC, the categories range from 1 to 5. This scale will be used in later analysis to delineate storms of various intensities. More on the sampling and storm metrics will be given in Section 3.1.

**Table 1.** Wind speed ranges within the Sapphir-Simpson hurricane intensity scale for classifying tropical cyclones (TCs).

| Sapphir-Simpson (S-S) | Range of Maximum Sustained Wind Speeds [m/s] |
|---|---|
| Tropical Depression (TD) | $\leq 17$ |
| Tropical Storm low (TS low) | 18–25 |
| Tropical Storm high (TS high) | 26–32 |
| Category 1 (Cat 1) | 33–42 |
| Category 2 (Cat 2) | 43–49 |
| Category 3 (Cat 3) | 50–57 |
| Category 4 (Cat 4) | 58–69 |
| Category 5 (Cat 5) | 70+ |

*2.2. Satellite Altimeter Data*

Currently, global sensing of wave height is only possible with satellite radar altimeters, however the repeat cycle—that is, the time interval between consecutive observations at the same location—ranges from ~month to ~week depending on the mission. In addition, only information about significant wave height ($H_s$), i.e., the mean of the $\frac{1}{3}$ highest waves, is available, and the quality of the estimate has been shown to deteriorate near the coast [33,34]. Ribal and Young, 2019 [33] flag data within 50 km of land and Dodet et al., 2020 [34] show poorer performance for buoys located within 200 km of the coast. While the along track resolution is ~7 km, the cross-track distance at the equator is an order of magnitude higher with shorter (longer) distances between sequential tracks for longer (shorter) repeat cycles.

Satellite-mounted radars send a pulse in the nadir direction, that spreads concentrically in space as it hits the Earth's surface. Geophysical characteristics can be derived from the waveform of the echo, or return signal. The waveform is a step function if the reflecting surface is flat, but if the surface is rough, the step function becomes skewed, with sloped leading and trailing edges. Significant wave height is related to the slope of the leading edge.

Radar altimeters have flown on polar orbit satellites since the 1970s. By the 1980s, the processing and collection of this data were high quality and considered operational. This study takes advantage of an altimeter dataset curated by [33] (RY19). RY19 quality controlled and inter-calibrated 13 altimeter missions, including GEOSAT, ERS-1, TOPEX, ERS-2, GFO, JASON-1, ENVISAT, JASON-2, CRYOSAT-2, SARAL, JASON-3, HY-2A, and SENTINEL-3A, altogether spanning 33 years, 1985–2018. We use their "adjusted" data that are a result of recalibrating each mission for better consistency across time (and missions). The altimeter data are served at 1 Hz. For each measurement, there is a quality control flag. Tamizi and Young, 2020 [22] used only the highest level of quality control (QC-1)-labeled "good data" in [33]-and they found the data too restrictive for analysis that binned spatial measurements by TC attributes. Similarly, we found that compared with the second level—(QC-2)-labeled "probably good data" in [33]—the density of observations of QC-1

restricted data was decimated everywhere around the storm (see Appendix A), therefore, we have opted to use QC-2 data.

Pairing Altimeter Observations with Storms and Storm-centered Reference Frame

A system of pairing altimeter data was developed within the auspices of the U.S. Army Corps of Engineers (USACE) Wave Information Study (WIS). The system is called altWIZ ([35] https://github.com/Tripphysicist/altWIZ.git, accessed on 1 January 2021). altWIZ can be used to pair RY19 data (or the data of the European Space Agency Sea State Climate Change Initiative [34]) with model results, stationary or drifting buoy data, or storm tracks. Using the storm track function, options were set to gather altimeter data within 500 km and 0.5 h of each TC coordinate. Altogether, there were 6709 altimeter passes resulting in 584,748 individual altimeter observations of within 500 km of North Atlantic LPS. For 1286 of these altimeter passes (119,989 observations), the LPS was a TC according to Sapphir-Simpson (S-S) intensity scale. Overall, 290,000 observations were within $20R$ (about a third was collected while the LPS was TC strength). Note, along with the track location (coordinates), the TC attributes of forward translation velocity $V$, maximum wind speed $U_r$, and radius of maximum winds $rmw$, if available, were paired to the altimeter data. This is done without further interpolation, as these parameters tend not to change significantly within each hour.

Throughout this study, results are presented in a storm-centered reference frame. Geographic coordinates are transformed to a storm-centered reference frame using the orientation of the storm and the $rmw$. Using the Haversine formula, the distance between observation coordinates to the storm coordinates is estimated. This distance is normalized by the $rmw$, $R = distance(observation, TC)/rmw$. The vector from the storm center to the measurement location is rotated by the storm orientation. Taking a cue from CBLAST, results are presented sectors divided by previously described angles with Cartesian coordinates that can be expressed by $R_y = R\sin\theta$, $R_x = R\cos\theta$.

### 2.3. Operational Hindcasts

We use model output from operational hindcasts. Only operational hindcasts have the length of record and basin-wide coverage needed to produce a very large set of paired data. In addition, this is an opportunity to evaluate how these hindcasts perform in TCs, and thus their utility for design engineering in areas where TC waves dominate the wave climate.

A hindcast usually consists of a number of telescoping grids that increase resolution near coastal areas where gradients increase and intermediate (and perhaps shallow) water physics become important. The hindcasts considered here use the 3rd generation (3G) spectral wave modeling system WAVEWATCHIII [36] (WW3). WW3 is a generalized modeling framework with multiple options for source term physics packages and numerics. The source term physics packages used include ST2 [37] or ST4 [38,39]. A detailed comparison of source term packages can be found for general cases in [40] and specifically for TCs in [41].

For evaluation, we only look at output from the largest (and lowest resolution) grid from each hindcast. The higher resolution grids are still important because the lower resolution grids use outputs that are upscaled from the higher resolution sub-domains. The evaluation statistics are nearly the same for the high resolution grids and the same domain within the low resolution grid (not shown). We require the altimeter observed $H_s$ to be at least 1 m, to avoid potentially noisy observations [34,42]. We use standard error statistics of *bias*, root-mean-square-error (*rmse*), percent bias or relative bias (*nbias*), normalized-rmse (*nrmse*), and correlation coefficient ($R^2$) defined as the following:

$$bias = \overline{H_s^{model}} - \overline{H_s^{obs}} \tag{1}$$

$$nbias = \frac{\overline{H_s^{model}} - \overline{H_s^{obs}}}{\overline{H_s^{obs}}} \qquad (2)$$

$$rmse = \sqrt{\overline{(H_s^{model} - H_s^{obs})^2}} \qquad (3)$$

$$nrmse = \sqrt{\frac{\overline{(H_s^{model} - H_s^{obs})^2}}{(\overline{H_s^{obs}})^2}} \qquad (4)$$

$$R^2 = \frac{\overline{(H_s^{model} - \overline{H_s^{model}})(H_s^{obs} - \overline{H_s^{obs}})}}{(\sqrt{\overline{H_s^{model} - \overline{H_s^{model}}})^2})(\sqrt{\overline{(H_s^{obs} - \overline{H_s^{obs}})^2}})} \qquad (5)$$

Generally, evaluations are only performed on data within a normalized distance of 20*R*. It is important to recall that the hindcasts use different grids to cover different areas over different time periods, thus they do not see the same storms. So, although one can quote numbers for each comparison, these cannot be compared directly against one another. Furthermore, these are not controlled comparisons, so interpreting differences between models is not straight forward. A summary of hindcasts characteristics can be found in Table 2.

**Table 2.** Hindcast models, grid resolutions, temporal output resolution, period covered, and boundary conditions.

| Grid/Hindcast | Model | Spatial Resolution | Time Output | Period Covered | Active Boundary |
|---|---|---|---|---|---|
| **WIS** | | | | | |
| North Atlantic Basin | WW3 (ST4) | 30 arcmin | 1 h | 1980–2019 | None |
| U.S. East Region | WW3 (ST4) | 15 arcmin | 1 h | 1980–2019 | N.A. Basin |
| U.S. East Coast | WW3 (ST4) | 5 arcmin | 1 h | 1980–2019 | U.S. East Region |
| **NCEP** | | | | | |
| Global | WW3 (ST2 + ST4) | 30 arcmin | 3 h | 2005–2019 | |
| North Atlantic Basin | WW3 (ST2 + ST4) | 10 arcmin | 3 h | 2005–2019 | Global |
| U.S. East Coast | WW3 (ST2 + ST4) | 4 arcmin | 3 h | 2005–2019 | N.A. Basin |
| **Ifremer** | | | | | |
| Global | WW3 (ST4) | 30 arcmin | 3 h | 1990–2016 | |
| North Atlantic Basin | WW3 (ST4) | 10 arcmin | 3 h | 1990–2016 | Global |

### 2.3.1. WIS

WIS is part of the USACE Engineer Research and Development Center's (ERDC) Coastal and Hydraulics Laboratory (CHL). WIS work has been ongoing since the 1970s, today's incarnation delivers wave climate information along all the U.S. coastlines, great lakes, and island territories [43].

WIS utilizes nested grids with refined resolution towards coastal areas. The Atlantic basin is covered by a 30-arcmin resolution grid that extends −84–20° E and 0–72° N, for which there are no global boundary conditions. This basin grid serves as active boundary conditions for a 15-arcmin resolution U.S. East Region grid and a 5-arcmin U.S. East Coast grid.

For all grids, the 3rd generation spectral wave model WAVEWATCH III® (WW3) [36] is used. The ST4 source terms were selected with default parameters and tunings. The model is forced by winds produced by Oceanweather Inc (OWI). OWI uses the statistical downscaling, data assimilation, and kinematic analysis [44] of select tropical and extra-tropical systems to enhance winds fields from [45]. WIS ouputs hourly field data from 1980 to 2018.

### 2.3.2. NCEP

The U.S. National Oceanic and Atmospheric Administration (NOAA) National Center for Environmental Prediction (NCEP) maintains several hindcasts. Here, we examine output from their multi-grid production hindcast. In contrast to the other hindcasts, this one is not suitable for climate application due to systematic model changes over time. This hindcast utilizes a nested-grid approach with a 30-arcmin resolution global grid, informing a 10-arcmin Atlantic grid, likewise informing a 4-arcmin grid that contours the east coast. The WW3 model is used with the ST2 source term package from 2005 to 2015, and ST4 from 2015 to 2019[4].

The models are forced by winds from NOAA's Global Forecast System[5]. We accessed monthly grib files on a public server that currently has model data from 2005 to the present (https://data.nodc.noaa.gov/thredds/catalog/ncep/nww3/catalog.html, accessed on 1 Novemeber 2020). The product has gridded wave height output every 3-h.

### 2.3.3. Ifremer

The Institut français de recherche pour l'exploitation de la mer (Ifremer) has run a global hindcast from 1990 to 2016. Nested grids increase in resolution towards coastal areas. They use the WW3 model with ST4 (TEST471) source term package forced by winds from the Climate Forecast System Reanalysis (CFSR) produced by NCEP [46–48]. The global grid has 30 arcminute spatial resolution. According to their documentation, there is no current coupling, no water level forcing, no ice forcing, but there is a source term for iceberg forcing for some years. North West Atlantic grid (data are missing from 1992–9 to 1992–12) has a spatial resolution of 10 arcminutes from 8–53° N and $-100°$ to $-43°$ W. For all models, there is a 3-hourly temporal resolution for gridded fields. Monthly field data files were accessed via the Ifremer FTP server (ftp://ftp.ifremer.fr/ifremer/ww3/HINDCAST/, accessed on 1 November 2020).

## 3. Results

### 3.1. Storm Attributes and Altimeter Observations

Within the TCs sampled by altimeters, there are a range of attributes. The three explored in this paper are the speed at which the storm translates through space (also referred to as forward velocity or translational velocity or forward tranlation speed), $V$, maximum wind speed, $U_r$, and radius of maximum winds, $rmw$. Of these, the primary driver of wave height is maximum wind speed. Forward velocity is important for producing dynamic fetch (e.g., [13,20]). $rmw$ is the key length scale and is used to normalize spatial data in the TC-centered frame of reference. Figure 1 shows the distribution of these 3 storm attributes associated with each altimeter pass (6709 passes).

Figure 1 shows a comprehensive sampling of storms with $V$ that range from 0–25 m/s, $U_r$ from 10–75 m/s, and $rmw$ 7–200 km. The distributions of $V$ are skewed, with the majority of sample $V \leq 10$ m/s. Likewise, the majority of TCs have $U_r \leq 40$ m/s. While the modal $rmw$ is around 50 km, there are at least 200 samples per bin within 7–150 km. For dynamic fetch, the joint distribution of $V$ and $U_r$ is informative.

Figure 2 is the joint distribution of TC maximum winds and TC forward velocity shows that very intense storms tend to move more slowly ∼5 m/s, whereas very fast moving storms, >15 m/s, tend to be either TS or Cat 1. Waves generated on the right side of the storm have increased development potential because of dynamic fetch, where there can be some correspondence between the wave group speed and direction; the forward translation speed and direction. Therefore, storms that provide these conditions would have properties somewhere on or to the right of line representing wave speeds in Figure 2. Above the line the storm would outrun the fastest waves still receiving energy from the winds and below the line the wave groups outrun the area of generating winds. To the left

---

4　See documentation at https://polar.ncep.noaa.gov/waves/hindcasts/prod-multi_1.php.

5　See https://www.emc.ncep.noaa.gov/emc/pages/numerical_forecast_systems/gfs.php.

of the line, the wave phase velocity is faster than the maximum wind (i.e., swell that is no longer growing due to input), to the right, the max winds are faster than wave phase speed (growing wind-seas). Indeed, ref. [20] has shown, theoretically, that a critical relationship between these two exists around $V/U_r = 0.36$.

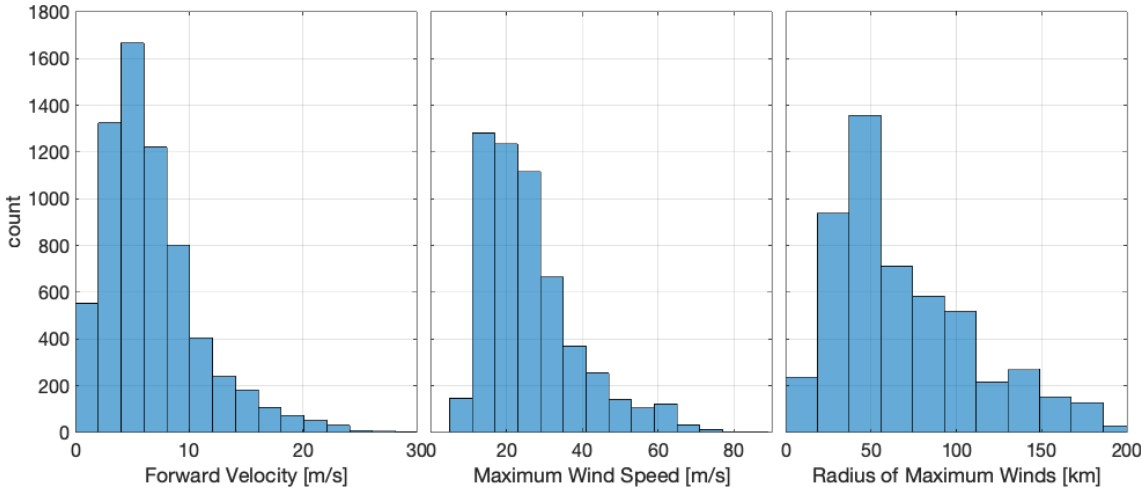

**Figure 1.** Histograms of (**left**) forward velocity, (**center**) maximum sustained wind speed, and (**right**) radius of maximum winds.

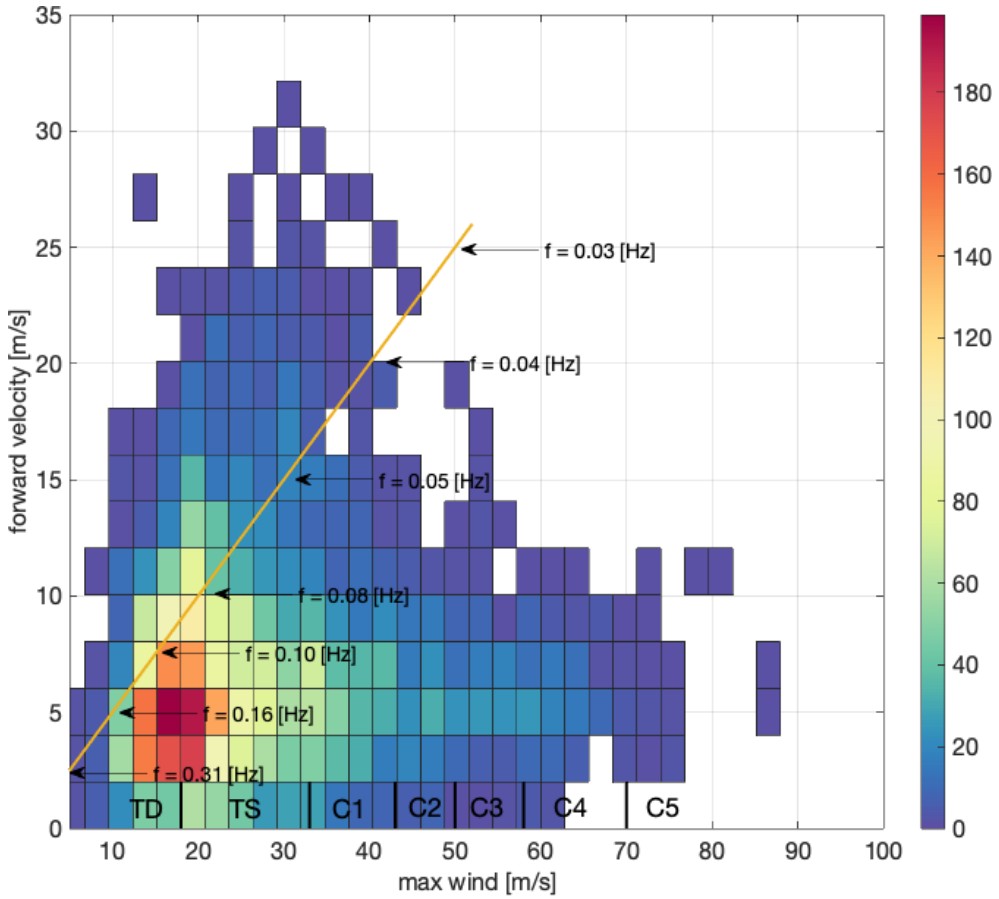

**Figure 2.** The joint distribution of maximum sustained winds and forward velocity; the counts are shown in the color scale. Additionally indicated are the S-S categories on the x-axis with black lines. An orange line shows the wave phase speed (celerity) that corresponds to wind speed on the x-axis and the wave group speed that corresponds to the translation velocity. A few frequencies in the wind-wave band are indicated along the line.

The wind field of a TC has a structure that can be parameterized in terms of a number of measurable variables, such as $V$ and $rmw$ (e.g., [10,11]). Because of this predictable structure, it is valid to visualize the observations from different storms as a composite. However, this ignores secondary structures within storms, such as eyewall replacement cycles and the concentration of high winds into bands and squalls; these secondary features have been found to affect wave generation [24].

In Figure 3, the average observed wave height is viewed in the TC-centered reference frame. The small circle represents the approximate location of the eye wall, $R = 1$, the larger circle is eight times this distance, $R = 8$, and the straight dashed lines break up the TC into 3 sectors: back, left, and right. Here, data have been restricted to storms with $U_r \geq 33$ m/s. The wave heights are asymmetric with the highest wave heights in the right sector around $R = 1$. The left to right asymmetry is pronounced, at $R = 5$, there is a $\sim$1.5 m difference between the wave height in the right and left sectors along the $nY = 0$ transect (Figure 3c). There is also a less pronounced front-to-back symmetry, with higher waves on average in the back hemisphere (Figure 3c).

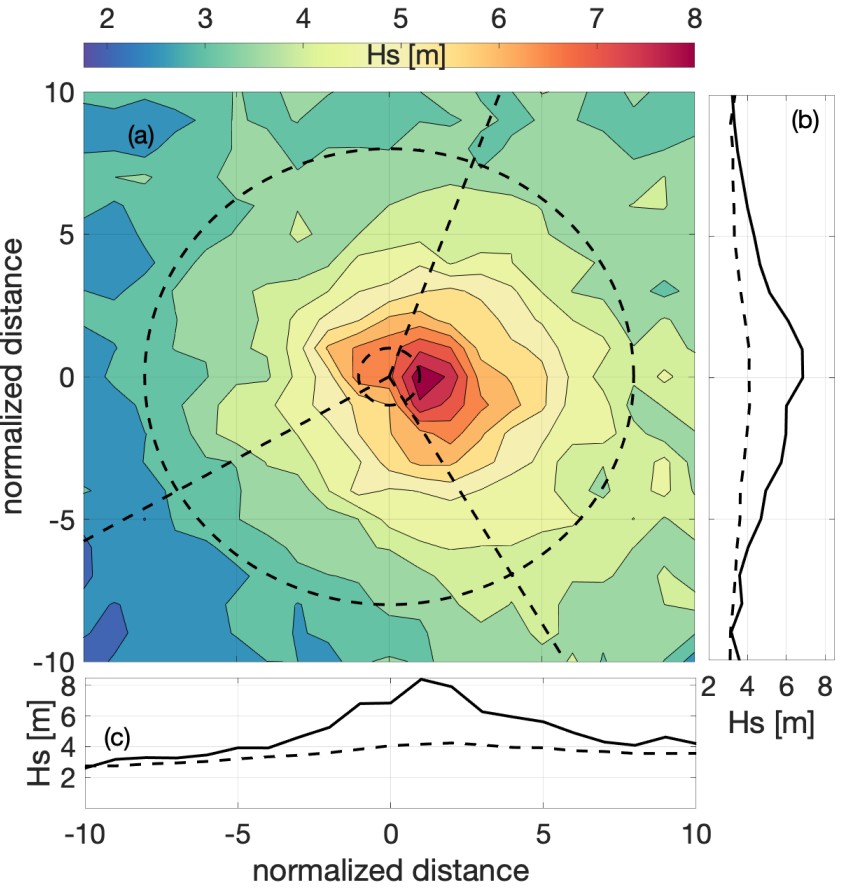

**Figure 3.** Average wave height observed by altimeters in a TC-centered reference frame ($U_r \geq 33$ m/s). (**a**) Color scale indicates average wave height observed in $2R \times 2R$ bins, with contour lines every 0.5 m. Small dashed circle is $R = 1$; larger dashed circle is R = 8. Dashed lines show TC sectors according to [27]. Up arrow is the orientation of the TC, moving forward up the page. (**b**) Solid line is a back to front transect along $nX = 0$ and the dashed line is the average along the nX axis. (**c**) as in (**b**) for nY.

Next, we parse wave height observations by maximum wind speed, following the S-S categories, and the translation speed. The subplots in Figure 4 are similar to Figure 3, but since some combinations of hurricane wind speed and translation velocity have fewer observations, we display the averages in each $2R \times 2R$—bin without smoothing between

bins. A minimum of 15 observations per bin was set. Thus, the subplots in Figure 4 are pixelated.

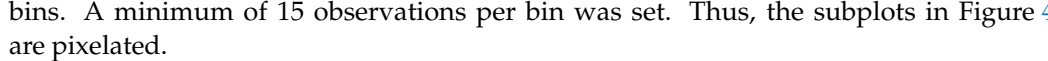

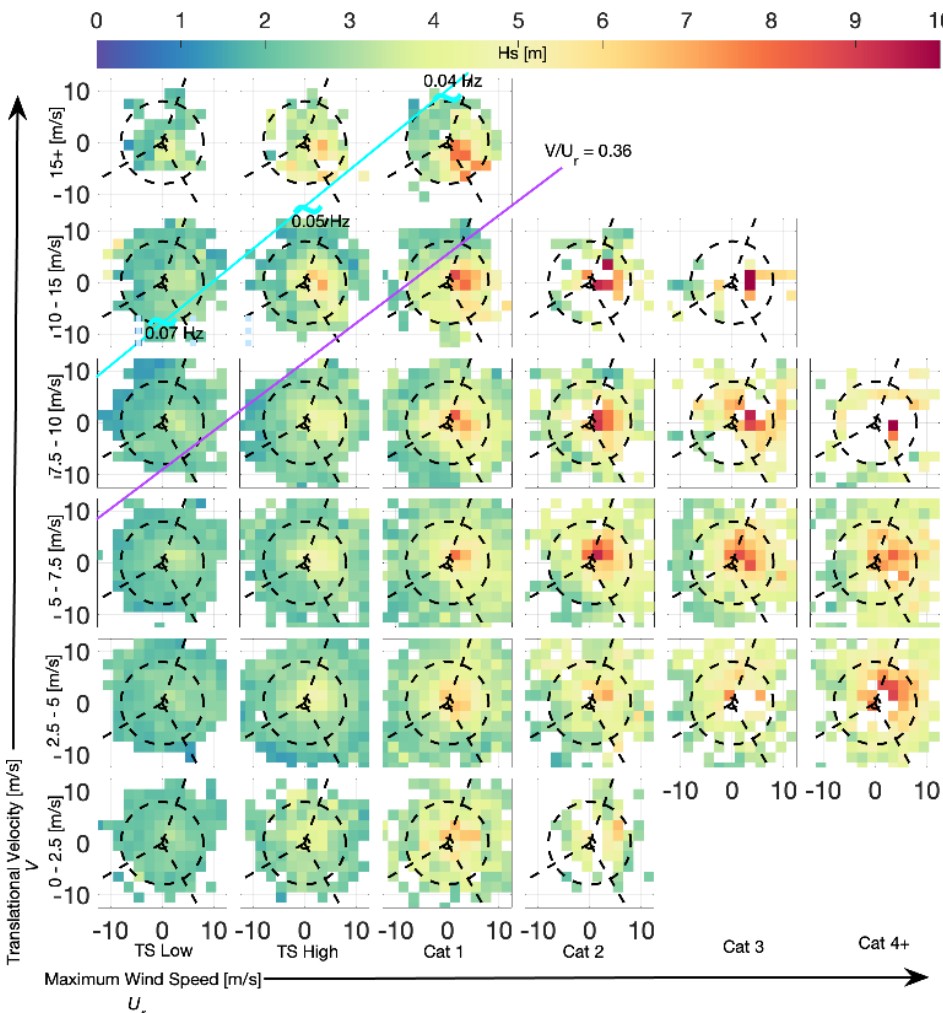

**Figure 4.** Altimeter observed wave height shown in color. Each subplot as in Figure 3 without interpolation or contour lines. Data are binned by S-S category from left to right. From bottom to top, data are binned by TC translation velocity. The color scale indicates significant wave height from 0–10 m. The cyan line shows the approximate correspondence of linear wave phase speed (x-axis) to group speed (y-axis). The wave frequency is given at 3 locations along the line. The purple line is the approximate location of the resonance condition of Kudryavtsev et al., 2015.

The x-axis shows maximum wind speed as represented by the S-S hurricane intensity scale. Because there is an abundance of tropical storms and a paucity of very intense storms, TS is split into low and high, and the highest two categories (4 and 5) are merged together. Moving left to right, for any row (constant translation velocity), the wave heights everywhere around a storm tend to increase. This follows directly from what is known of wave generation; wind sea wave heights are proportional to the square of the wind speed, faster winds result in larger waves. Moving bottom to top, instead of wave height increasing across the board, the increases are concentrated in the right sector, in other words, the largest waves get larger. This is particularly evident for the TS high and Cat 1, where the full range of storm speeds is captured, and in Cat 2, where the pattern emerges but there simply are not enough fast moving Cat 2 storms to fill out the plots. Cat 3 and 4+ storms show less variation of wave height as a function of storm speed, and apparently, no fast translating Cat 3+ storms were sampled. Wave energy in the wind wave band propagates from 1.5–16 m/s (periods of 2–20 s), thus forward storm movement in this

range increases the effective fetch. Since only waves in the right sector benefit from the translation velocity, the asymmetry of the wave height increases. Note also that as the translation velocity increases, the location of the peak wave heights shifts from ahead of the eye to behind.

### 3.2. Hindcast Evaluation-Overall Statistics

Here, we present all statistics calculated from all model-observation pairs within 20*R* and 10*R* of an LPS, regardless of storm attributes. Figure 5 shows the statistics for each of the hindcast grids considered as a function of wind speed at the center of the LPS. To put these numbers into a more general context, we can consider statistics from hindcasts evaluations that were not resitricted to LPSs [34,35,49]. WIS was evaluated in [35]. For the Atlantic Basin grid between 15° and 60° N for all of 2017 and found $R^2 = 0.90$, bias $= -0.07$ m, and rmse $= 0.43$ m. Ref. [34] evaluated IFREMER North Atlantic grid from 1994–2018 and found $R^2 = 0.91$, bias $= 0.04$ m, and rmse $= 0.31$ m (nrmse $\sim$12%). Ref. [49] evaluated NCEP 30-year homogenous hindcast (phase-1) global grid with a sliding 3-month window; from 2005–2010 $R^2$ ranged from 0.90–0.95, bias ranged from 0.15–0.35 m, and rmse ranged from 0.43–0.58 m. Keep in mind that the details vary considerably when it comes to model setups and methods of calculating error statistics (for full details, please see the original sources [34,35,49]).

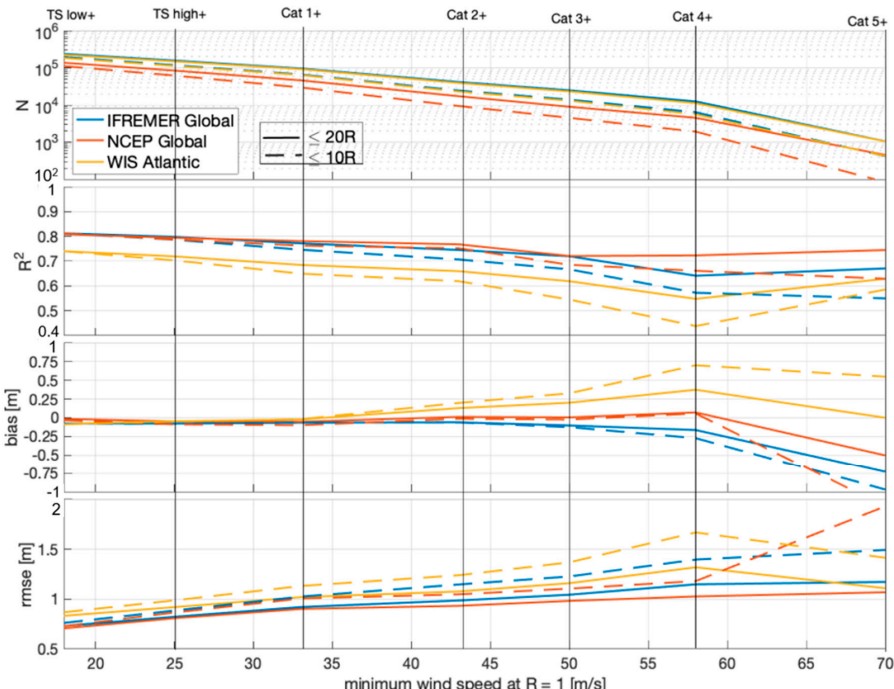

**Figure 5.** Error statistics for each operational hindcast, Ifremer in blue, NCEP in red, and WIS in yellow. Solid lines are for data within 20R of an LPS, and dashed lines are for data within 10R of an LPS. The x-axis is the minimum required wind speed of the LPS. S-S status is indicated by vertical lines. From top to bottom is the number of comparison, correlation coefficient, bias, and rmse.

Compared to the figures quoted above, even for the most inclusive statistics during LPSs (solid lines on left side of Figure 5), there is decreased $R^2$, decreased *bias* (from positive to negative overall), and increased *rmse*. Below TC level, hindcast results are similar between IFREMER and NCEP, WIS has decreased $R^2$ and increased *rmse*. As the wind speed requirement increases, the amount of data compared decreases exponentially and statistics tend to degrade. $R^2$ tends to decrease while *rmse* tends to increase, and a spread between hindcasts becomes evident. For comparisons that include data lower than Cat 1 wind speeds, all models have a negative bias. For Ifremer, the negative bias stays

negative, decreasing to −0.10 m and −0.16 m for Cat 3+ and Cat 4+ storms. For NCEP bias is closest to 0 m, going positive for Cat 2+ and maxing out a 0.07 m for Cat 4+ storms. WIS starts the lowest, −0.08 m for TS low+, but ends up with the highest bias of 0.38 m for Cat 4+. Limiting the comparison to data closer to the center of the LPSs (from 20R to 10R) tends to exacerbate the errors.

There are at least 2 factors contributing to the increased error for hindcasts data near LPSs: (1) average wave height sampled is larger resulting in larger absolute errors; (2) actual increase in disagreement in which the models are not as skillful in producing TC wave fields.

In the following sections, we further explore space and time dependencies of errors around TCs, and for the sake of brevity, we consider only the IFREMER Global hindcast. However, some additional analysis on NCEP and WIS are presented in Appendix A, and a detailed look at WIS is planned as future work.

### 3.3. Ifremer Global

Here, we present normalized error metrics—relative bias and nrmse. Unless explicitly stated, data are restricted to model-observation pairs within 20R of a TC with $U_r \geq 33$ m/s with coincident estimates of $V$ and $rmw$. For binned metrics, we require at least 15 observations per bin to calculate statistics.

### 3.3.1. Error with Wave Height

Figure 6 shows percentage bias and nrmse binned by observed wave height, bin averages and standard deviations are shown with solid lines and shaded regions, respectively. The lower plot is a histogram indicating the number of observations in each bin. Except for the first bin, on average, the percent bias is near zero, but decays steadily as a function of wave height, until about $H_s$ of 13 m, where it falls more precipitously (also where the number of samples is greatly reduced). The nrmse stays near ∼16% across the wave height bins until 13 m, where it increases drastically. Even discounting the accuracy of observed wave heights above 13 m, there is a clear tendency for underestimation by the model to increase as the observed wave heights increase.

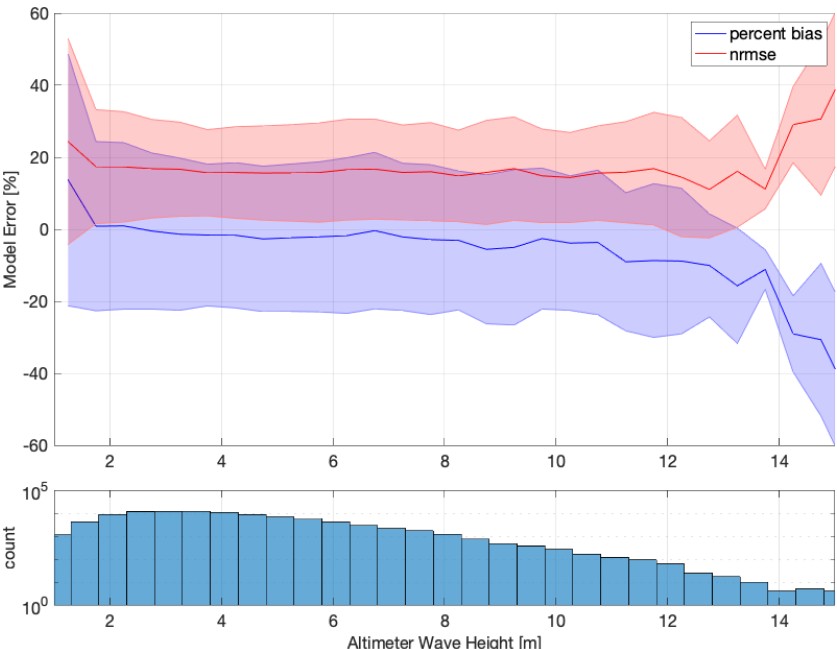

**Figure 6. Top**: percent bias and nrmse in blue and red as a function of observed wave height in meters. Solid lines are the averages and the shaded areas are the std within each bin, bin widths are 1 m. **Bottom**: histogram of samples within each wave height bin; counts shown on a log scale.

### 3.3.2. Error over Time

Given the time period over which data are available, there is an opportunity to explore error metrics over time. Figure 7 is similar to Figure 6 with error binned over time. The nrmse vacillates around 15% over time, with the exception of 2015 where it goes above 20% (likely related to a slow TC season with relatively few storms to sample). While variation of percent bias shows that there is both under and over estimation every year, on average, percent bias shows model underestimation for years 1992–2008, and overestimation from 2008–2016 maxing out at 20% positive bias in 2015.

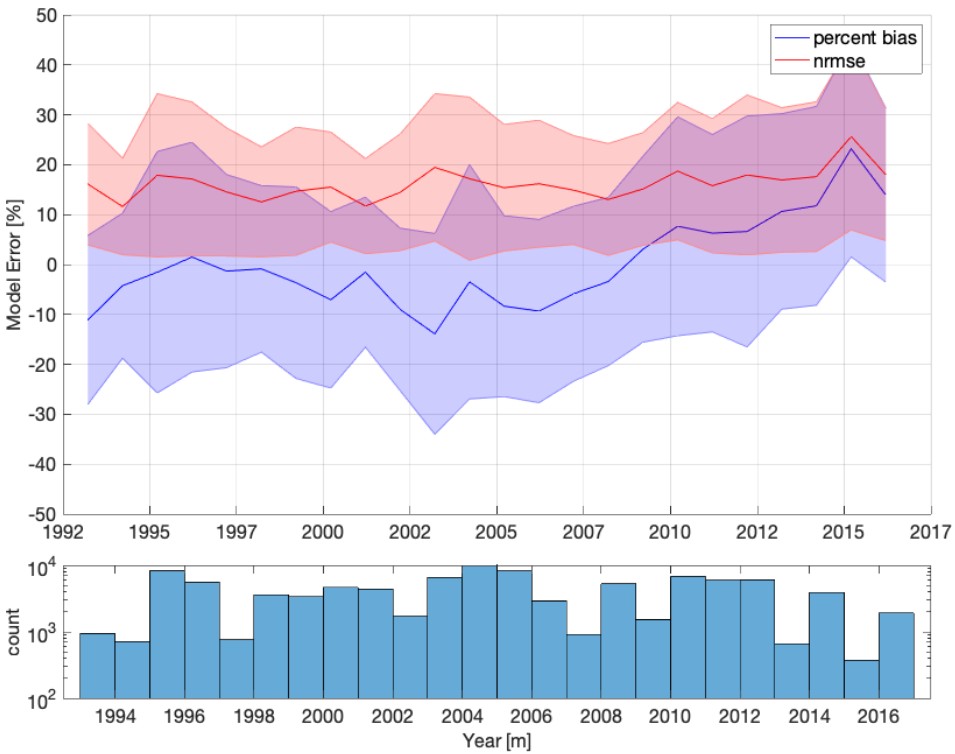

**Figure 7.** As in Figure 6 over time indicated in years with 1 year bins.

### 3.3.3. Error in Geographic Space
Geographic Observation Density

Similar to Figures 6 and 7, percentage bias and nrmse can be binned as a function of geographic space: latitude and longitude. To produce a map, we collect data into $1.5° \times 1.5°$ bins. In the plot, there is some interpolation of color shading between bins to give a smoothed appearance. First, we look at the concentration of observations to get an idea of where this analysis is valid, and where our confidence in the statistics are eroded due to insufficient sample size.

Figure 8a shows the number of observations per bin. There is a high concentration of observations in the geographic area where TC tracks often re-occur. There is a decrease in observation density near coasts because there is a corresponding decrease in data quality, and thus a decrease in passable observations. The areas of high obsveration density are consistent with typical TC pathways: easterly waves propagate off the west coast of Africa and develop into TCs in the low latitudes (10–20° N) [3]. A TC can track through the Caribbean and into the Gulf of Mexico or begin to gently curve north. If they curve towards the north, they eventually transition to a mid-latitude or extra-tropical cyclone, usually in the vicinity of 30–60° N [3]. A TC either makes landfall and dissipates or remains offshore and begins to propagate further north or northwest. This is backed up by Figure 8b that shows the location and velocity vector for each LPS considered with intensity indicated with color.

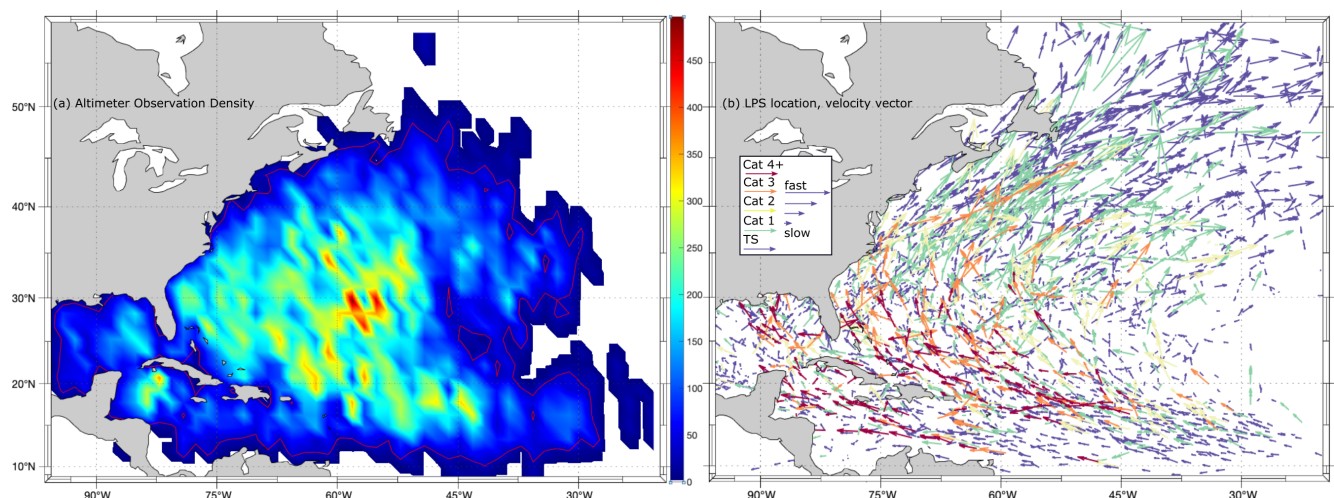

**Figure 8.** (**a**) Density of observations shown in color in 1.5° × 1.5° bins for the western portion of the North Atlantic basin. A thin red line shows the 15 observation per bin contour. (**b**) is the geographic location of each LPS coordinate. The translation velocity and direction is indicated with a vector arrow. The color indicates the S-S intensity.

Geographic Error Maps

Figure 9 shows relative bias and nrmse on a geographic map. There is a strong signal of underestimation in the low latitudes (and correspondingly high rnmse). This is the typical pathway of very intense and compact storms that move relatively slow through the region (see Figure 8b. In general, the map margins seem to be areas of higher highs and lower lows for relative bias, and higher nrmse, all of these corresponding to areas with relatively fewer observations. A notable exception to this is a region of over estimation around 33° N, 70° W, within an area where observations are relatively plentiful (as indicated by Figure 8a). This high relative bias is accompanied by a more localized point of high nrmse (Figure 9). There is some indication of the influence of coastal areas in the measurements, particularly with areas of high relative bias and *nrmse* in the Caribbean. There appears to be lower relative bias in lower latitudes, but there are no clear patterns otherwise. In particular, a pattern of increased variance in wave height has been observed in altimeter data over strong currents, including over the Gulf Stream and its extension [50]. This increased variance leads to elevated random errors for models that do not include wave–current interaction [35]. There is no indication of a Gulf Stream signal; either the resolution is too coarse or other factors dominate the error signals around TCs.

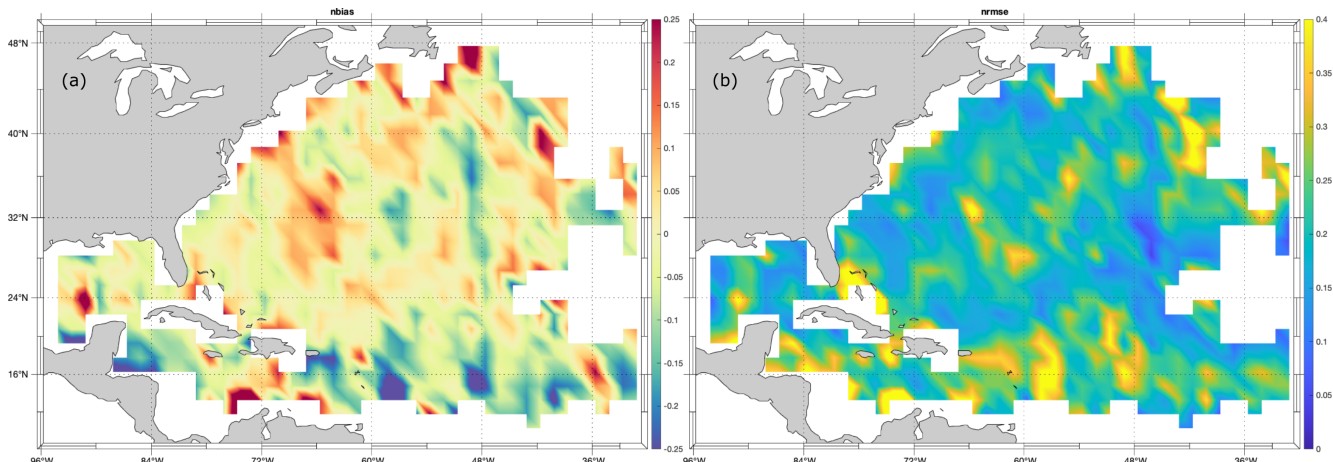

**Figure 9.** As in Figure 8a for relative (**a**) bias and (**b**) nrmse.

Figure 10 shows the percent bias and nrmse binned into 2° latitude bins (note that because of spherical geometry, the area considered is smaller as latitude increases). There are some noteworthy signals as a function of latitude: on average, there is slight underestimation from 6–10° N, transitioning to underestimation of wave height from 12–18° N, little bias from 20–40° N, and increasing over estimation from 42–52° N. The low is about −10% around 16° N and the (reliable) high of 10% around 48° N (above mid 40°, where TCs transition from tropical to extra-tropical cyclones and increase translation velocity). The rmse starts low and increases (reflecting the increased negative bias), then levels out in the 15% range before increasing as a function of latitude from 30° N to 50° N. On the whole, errors are less from 20–35° N.

Figure 11 shows bias and nrmse over 2° longitudinal bins. There is a spike in bias accompanied with a spike of nrmse near −95° W, which is in the Gulf of Mexico. The next nrmse spike is associated with the area around the Bahamas. This is a relatively shallow, island-rich region, and it is expected that disagreement here is partly due to poor performance of the altimeter near land and partly due to poor representation of the shadowing, blocking, and diffraction by the islands in the model (e.g., see Rogowski et al., this issue). The comparison is more stable as a function of longitude from −75° W to −30° W; the nrmse is around 15% and the relative bias is within ±8%. At the extreme eastern part of the basin, there is increased disagreement, but there are also fewer samples, as TCs tend to develop west of this area.

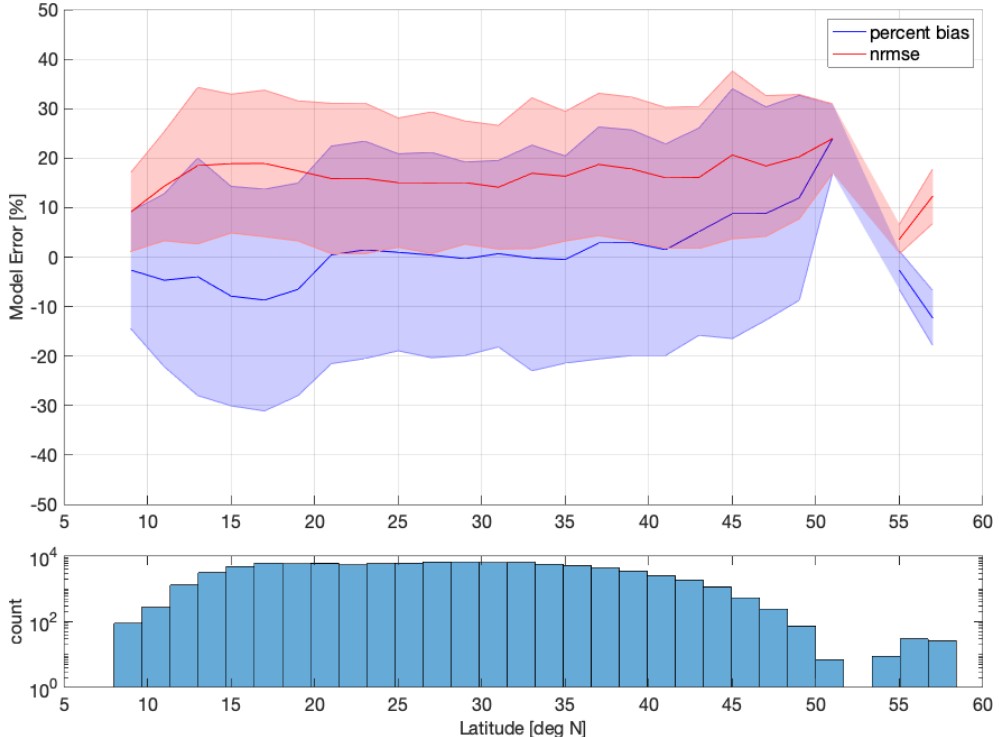

**Figure 10.** As in Figure 6 over 2° latitude bins.

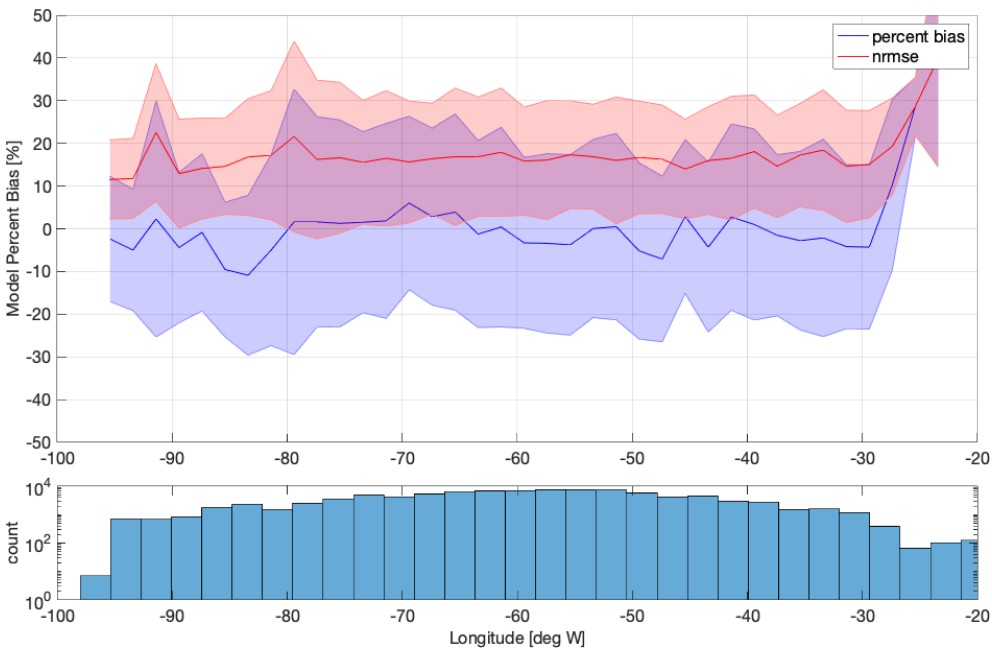

**Figure 11.** As in Figure 6 over 2° longitude bins.

### 3.3.4. TC Centered Reference Frame

In the following sections, errors are viewed in a TC-centered reference frame (as a reminder, the plots are set up for a TC in the northern hemispheres moving up the page). Generally, the observation density is a maximum near the center of the eye and decreases out radially (see Appendix A). The statistics are calculated for $2.5R \times 2.5R$ bins around the TC. The values between adjacent bins are blended.

Figure 12 shows the spatial patterns of relative bias and nrmse around a TC. There is general pattern of underestimation of wave height in the left sector ($\sim-4\%$) and overestimation of wave height in the right sector ($\sim+5\%$). The exceptions are directly ahead of the storm were technically in the left sector we observed slight over estimation and near the eye where wave height remains underestimated ($\sim-2\%$). There is a transition area ahead and behind the eye. In the back sector there are alternating areas of weak under and over estimation. These sectors, as suggested by [27] and others, grossly separate areas of over and underestimation within $R < 8$, but there remains nuance.

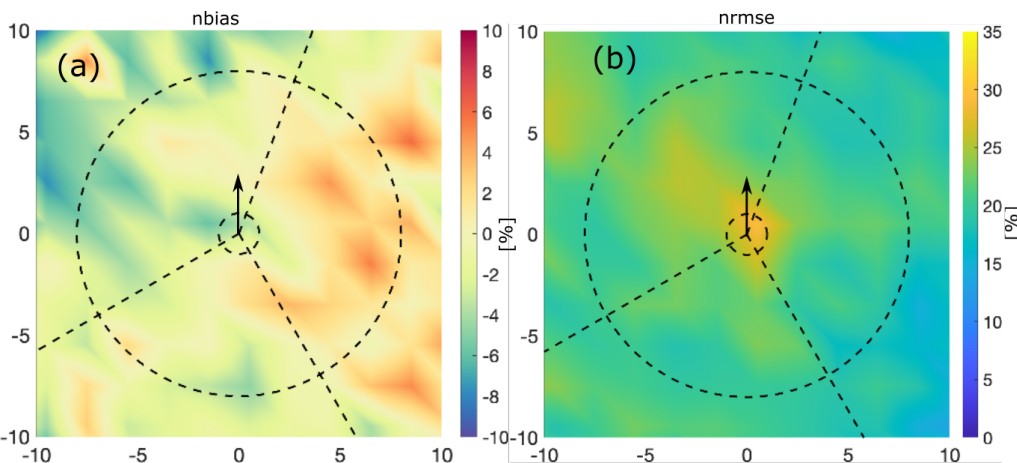

**Figure 12.** As in Figure 3 for relative nbias (**a**) and nrmse (**b**).

The nrmse is relatively high near the eye of the storm (30%) and decreases out radially. The right sector shows less nrmse than the left. High nrmse emerges near the margins ($\sim R = 15$, not shown). The observation density is very high within $R < 8$. $R = 8$

corresponds to the 600 samples per bin contour, but there are 100 or fewer samples per bin at the margins at 15R. Next, we parse model-observation pairs in terms of storm attributes: maximum wind speed, translation velocity, and radius of maximum winds.

TC Intensity

Next, we parse model-observation pairs in terms of storm attributes: maximum wind speed, translation velocity, and radius of maximum winds. First, Data were binned by intensity using the S-S hurricane intensity scale (refer to Table 1). Figure 13 shows relative bias for each intensity category. For TS Low through Cat 2 wind speeds, the patterns within $R = 8$ are similar to those found in Figure 12a. While the pattern is barely discernible for TS Low/High, it appears to strengthen with increasing intensity. For Cat 3 wind speeds, overestimation appears in all sectors except for the region around the eye and areas in the left sector. For Cat 4+ wind speeds, there is strong underestimation ($>25+\%$) within $R = 5$, mainly in the right sector peaking at $-35\%$. Outside of $R = 5$, the overestimation emerges in the right and back sectors before $R = 8$. At this most extreme bin, generally there is an overestimation behind the eye and underestimation ahead of the eye.

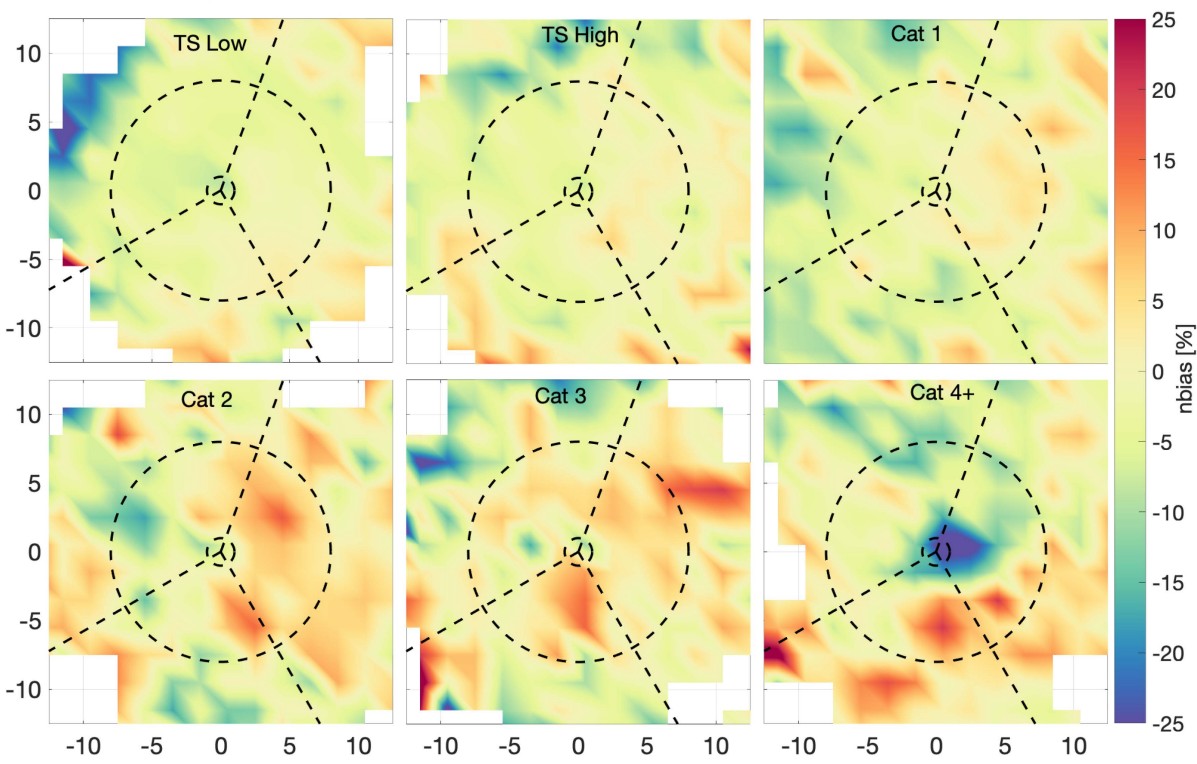

**Figure 13.** Each subplot as in Figure 12a for each category of the S-S intensity scale. (Note the difference in color scale).

Comparing subplots outside of $R = 8$, there is less consistency in relative bias across intensity, with overestimation found in the forward right of Cat 1 storms, and underestimation found in the left of the back section for Cat 3.

Figure 14 shows the nrmse parsed by intensity. The pattern in Figure 12b is also present, more or less, in each subplot of Figure 14. Within $R = 8$, the nrmse is high towards the center and lower as a function normalized radial distance, however, there are isolated areas of increased nrmse outside of $R = 8$. The nrmse tends to increase as a function of intensity as nrmse of 25% near the eye of TS low and 50% near the eye of Cat 4+.

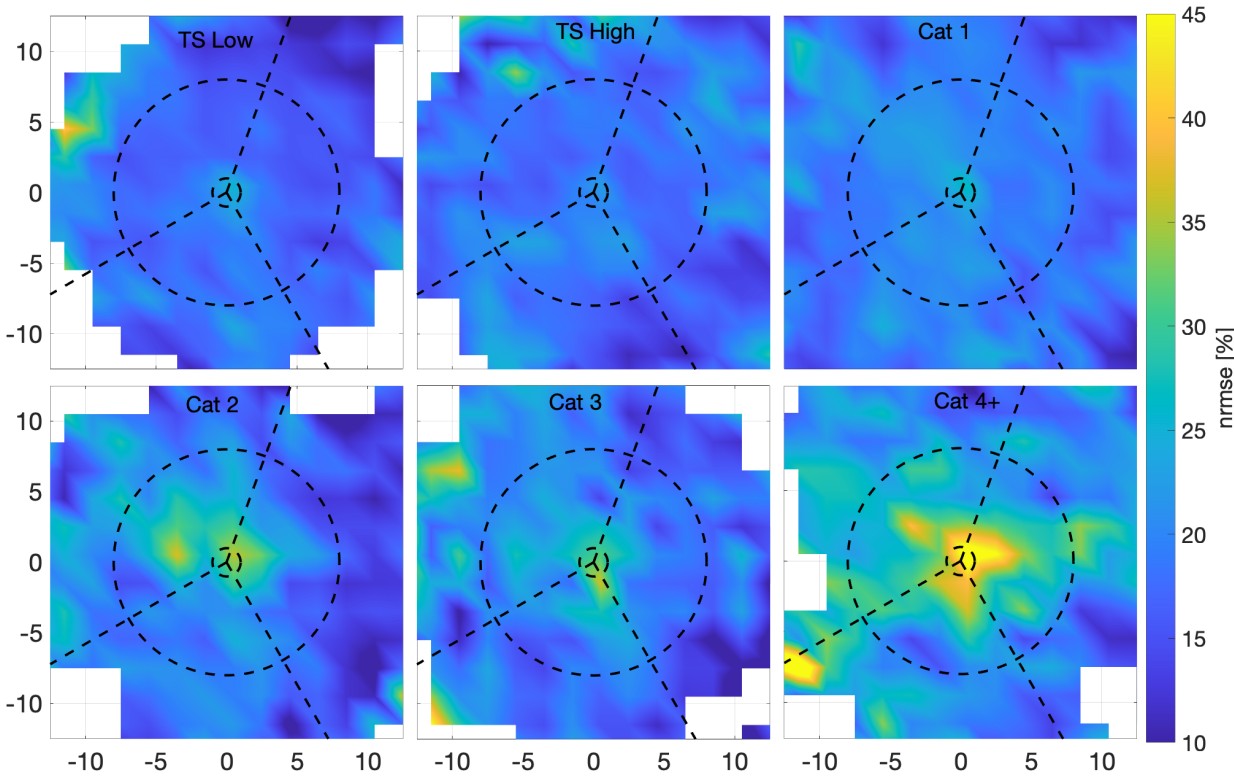

**Figure 14.** Each subplot as in Figure 12b for each category of the S-S intensity scale (note the difference in color scale).

TC Forward Translation Velocity

Next, we set $U_r \geq 33$ m/s (Cat 1) and bin the data in terms of forward translation velocity. Figure 15 shows relative bias as a function of TC translation velocity. The pattern identified in Figure 12a is also present in these subplots. For all subplots, there is a relative minimum in bias around the eye, and this bias is negative (model underestimation) for all but the fastest storms. There is a systematic shift of relative bias as a function of $V$. For slow moving storms, $V = 0$–2.5 m/s, there is a tendency for under estimation (10%) in left and back sectors nearly transitioning to positive bias in the right sector. For moderately fast storms, $V = 7.5$–10 m/s, there is slight negative bias in the left sector slight positive bias in right and back sectors. For fast storms, $V = 10 - 15$ m/s, there is positive bias all sectors and it is relatively strong in the right sector (10%).

Figure 16 shows nrmse for the same forward translation velocity bins. There is nrmse local maxima within $R = 8$, but it is slight. These are subtle shifts, but no strong pattern emerges in nrmse as the storm speed increases.

TC Radius of Maximum Winds

Finally, model-observation pairs are parsed by *rmw*, the defacto TC length scale. Figure 17 shows relative bias for storms with increasing *rmw*. For the smallest storms, *rmw* $\leq 30$ km, there is severe underestimation ($\sim$30%) around the eye in all sectors. There is marked negative bias in the left sector, and positive bias in the right sector outside closer to $R = 8$. Outside $R = 8$, overestimation emerges in the right and back sectors. For *rmw* = 35–45 km, there is underestimation (25%) around the eye in all sectors $R < 2$. This transitions to overestimation in the right and back sectors. The underestimation around the eye is reduced to ($\sim$15%) for *rmw* = 45–65 km, positive bias is also reduced. For larger storms, there is less data beyond a certain radius; this is simply because we started with data in an absolute frame of reference within 500 km of a TC. Thus, for a TC with *rmw* $\geq 80$ km, there are no data beyond $R = 5$. For the larger storms, *rmw* $> 60$ km, underestimation is much less prominent.

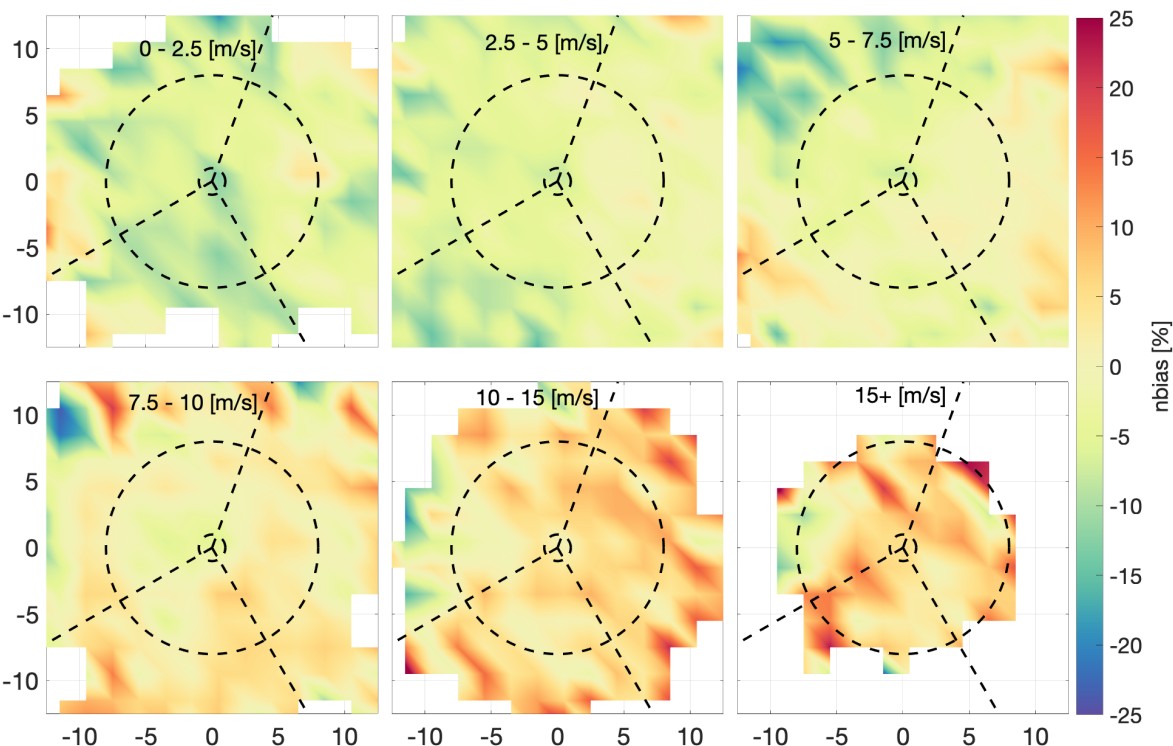

**Figure 15.** As in Figure 13, each subplot parsed by forward translation velocity.

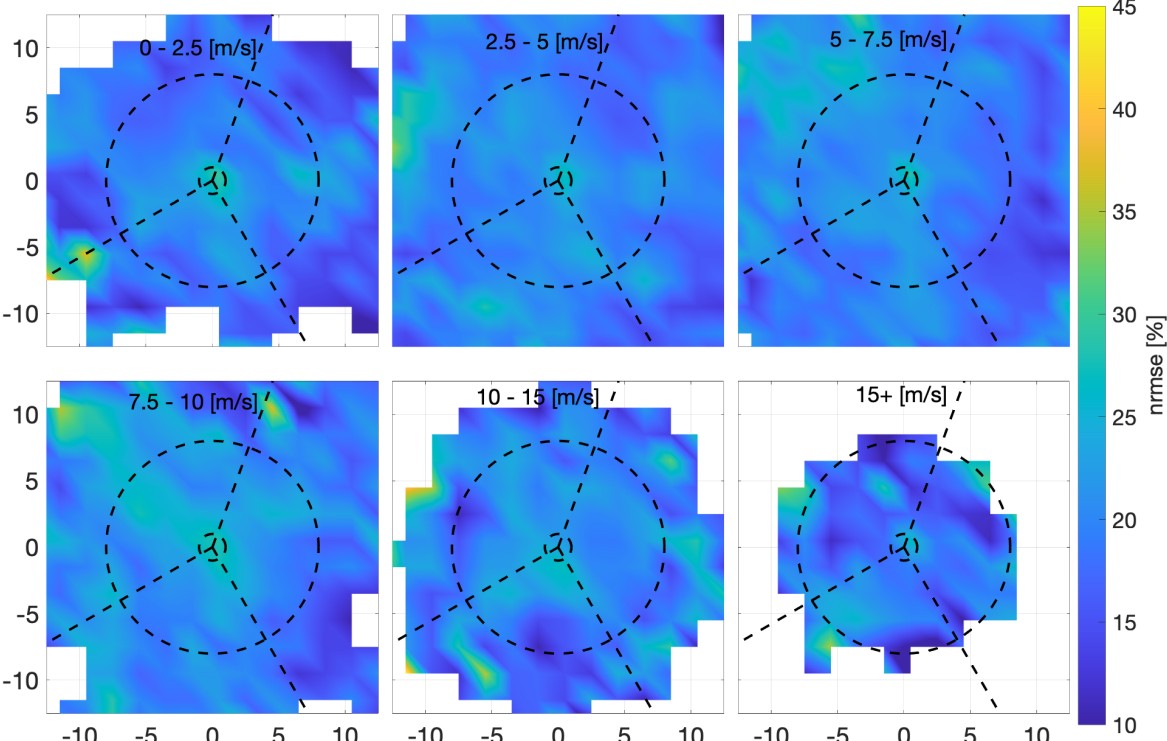

**Figure 16.** As in Figure 14, each subplot parsed by forward translation velocity.

Figure 18 shows nrmse for various *rmw* bins. For the smallest storms, *rmw* ≤ 30 km, there is very high nrmse (∼50%) around the eye and high levels everywhere within *R* = 8. For the next *rmw* bin, *rmw* = 30–45 km, nrmse is milder (40%) and more confined. The trend continues as *rmw* increases.

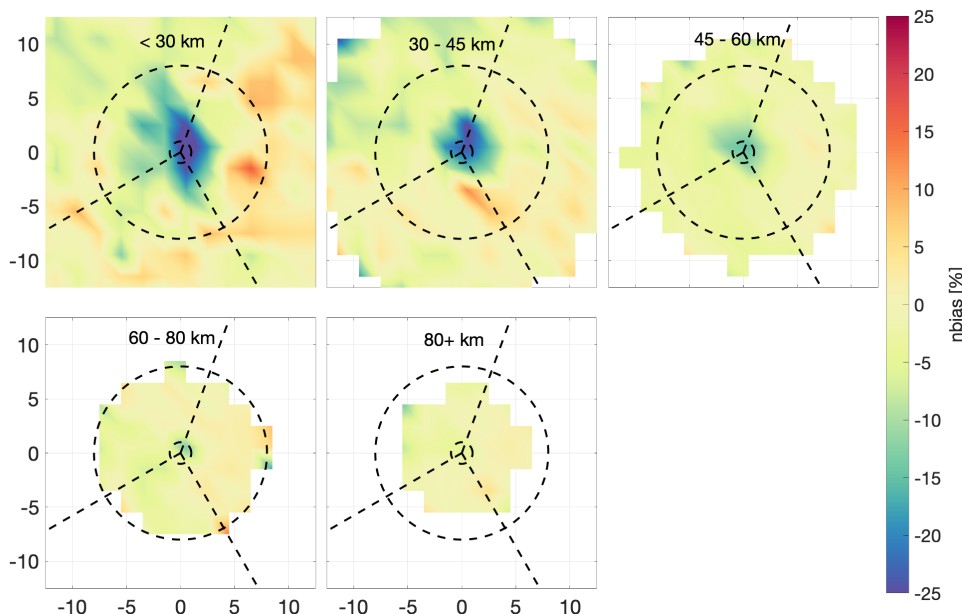

**Figure 17.** As in Figure 13, each subplot parsed by radius of maximum winds.

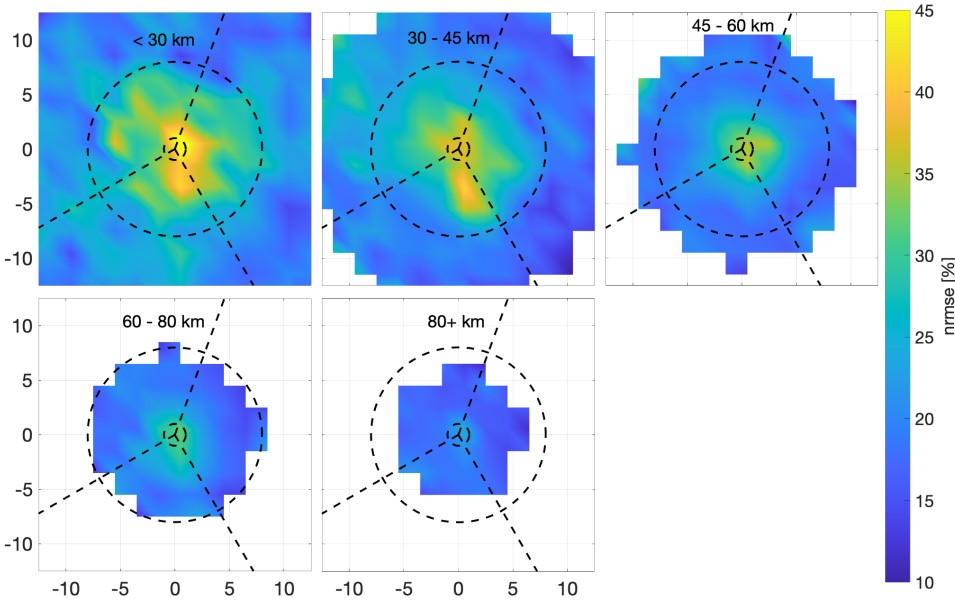

**Figure 18.** As in Figure 14, each subplot parsed by radius of maximum winds.

## 4. Discussion

Waves grow under the influence of wind. Three factors contribute to wave growth: fetch, duration, and wind speed (e.g., [51]). Compared to other weather systems, even ones that generate large waves such as extra-tropical lows in the North Atlantic [52], the wind structure around a TC is compact and it evolves quickly in time and space. The wind gradients imply that waves in a TC are typically fetch- or duration- limited. Duration and fetches are less limited (and perhaps lifted) for certain wave-frequency bands under dynamic fetch scenarios, when the wave group speed and direction matches the forward velocity and direction of propagation of the storm for substantial periods of time (e.g., [20]). Dynamic fetch qualitatively explains why wave height increase and asymmetry increases as a function of forward velocity in Figure 4.

For Figures 2 and 4, the wind speed along the x-axis is that which corresponds to the maximum wind located at the *rmw*. The energy in the wind-wave band is typically

above 0.05 Hz, thus all wind-waves are young near the *rmw* in a Cat 1 or higher storm. In a Cat 3 storm, the TC strength winds extend out to some multiple of *R*; this increases the instantaneous fetch for all waves ($H_s$ increases broadly space from left to right in Figure 4). As $U_r$ and the size of *rmw* increase, there is an increasing instantaneous fetch in which waves can potentially benefit from dynamic fetch, as reproduced by the parametric model of [13]. As waves grow, wave lengths become longer and phase and group speeds become faster (due to downshifting and 4-wave interaction). As waves grow longer and faster, they require increasingly large fetches to continue to grow. Since static TC fetches are small, very large waves depend on storm velocity to grow [20]. We observe that the peak wave heights shift from ahead of the storm to behind the storm as the storm translation speed exceeds the wave group velocity.

By performing model observation comparisons, we have implicitly assumed that altimeter data are of sufficient quality to treat as ground truth. However, it is appropriate to question this assumption. One issue is uncertainty in extreme waves as observed by altimeters: do the algorithms (and calibrations) remain accurate in conditions for which there is little buoy data for validation? RY19 and [34] show comparisons to buoy data with very little bias up to $H_s$ of 9 m. Although TCs generate very large waves, the average significant wave height from observations within 20R of Cat 1 and higher TCs was ∼4 m; when binned spatially, the highest bin in the right sector near the eye was ∼7 m (Figure 3). Indeed, only 1.5% of the data considered was greater than 9 m $H_s$. Another issue is that the spatial resolution as the altimeter has a footprint on order ∼7 km. The concern is that observations could blur the true wave heights and wind speeds, especially in areas of high gradients such as near the eye wall. This issue is mitigated by aggregating data from many different storms into composite plots ∼$2R \times 2R$ resolution, which is typically much larger than the measurement footprint. Even the smallest TCs have *rmw* > 7 km, and the very small TCs make up a small percentage of the population investigated. For the same parameters described above, there were no cases of *rmw* < 10 km, 0.4% with *rmw* < 15 km, and 6.0% of *rmw* < 20 km. For analysis and ground truth for comparison, we used the RY19 dataset. We chose to present results based on RY19 QC-2 data. Similar results were achieved with either the RY19 QC-1 or ESA data Appendix A. Potentially, there is an increase in rmse that is not associated with model performance. Additional implications of including lower quality data in the analysis are not well known.

Using TC information introduces uncertainty in the storm location. In IBTrACS, the storm coordinates are quoted to accurate to within 10–15 km [31,32]. Both TC-OBS and IBTrACS track storm attributes if they transition to extra-tropical. Therefore, the analysis will contain some samples from extra-tropical cyclones, though their population in the database is expected to be small given the observation density in the mid to high latitudes in Figure 8.

Multiple approaches are needed for evaluating and improving models. Satellite altimeters and stationary buoy networks provide complementary perspectives, and both are useful for understanding where and when models err simulating TC waves, respectively. For any individual storm, altimeter observations are sparse, so its utility for evaluating performance for any single storm is limited (e.g., [53]). However, the power of altimeter data is the ability to derive a model performance climatology. In this respect, altimeter data are complementary to data collected by coastal wave buoys (e.g., Rogowski et al., this issue). Buoy data are concentrated along the coastal areas (where the altimeter data tends to deteriorate), buoys provide detailed spectral and directional-spectral data (where altimeters only give significant wave height), and buoys report data from a single point on Earth over the life cycle of a storm. In a TC-centered frame of reference, buoy data is a transect (over the course of many hours) that reflects the evolution of the storm (i.e., the arrival of forerunners, etc.), where an altimeter pass is a purely spatial transect, practically instantaneous on the scale of storm development.

The spatial pattern of error presented here was derived from a composite of many storms, as such we do not expect this pattern to hold for every storm examined. Indeed,

Rogowski et al. (this issue) show that while there is a trend towards underestimation of peak Hs values, bias patterns vary from storm to storm. Because of the snapshot-like data, altimeters cannot address the time dimension implicit in the "missing the peaks" conundrum, but our main result is complimentary: that the highest waves around a TC (around the eye region) tend to be systematically underestimated. Furthermore, the higher the significant wave height, the more the model tends to underestimate; note this issue is not unique to TCs, e.g., Figure 16 of [54].

Figure 7 showed a relative bias signal with time. One possibility is an artifact of imperfect calibration between missions and changes in orbits (as described shown in [34] their Figure 11). Figure 19 shows relative bias over time, similar to Figure 7, but with the addition of WIS and NCEP. Note that the NCEP model was not designed for looking at trends over time, as there are systematic changes within the model including source terms. We find that NCEP shows a similar error trend from 2006–2014, but WIS is relatively flat over the same time period. To some degree, this stems from model forcing, OWI in WIS, CFSR in IFREMER, and GFS in NCEP. There is evidence of systematic changes of CFSR over time [55], particularly with the implementation of improved data assimilation and resolution following 2011 [48]. Both changes in wind fields and source terms may introduce different error signals over time. Whatever the source, bias in NCEP and IFREMER has non-trivially increased from 2006–2014 during TCs.

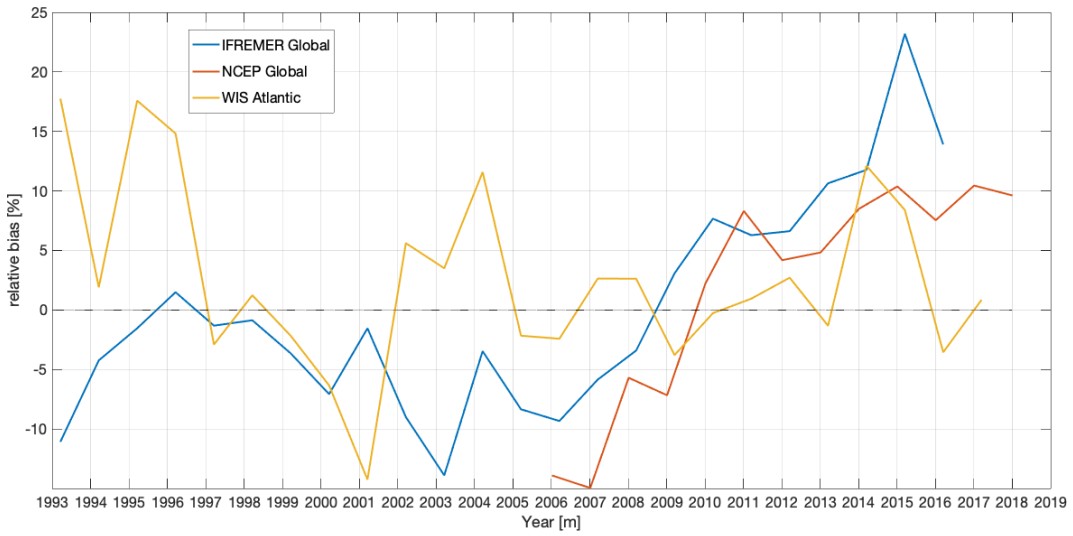

**Figure 19.** Relative bias binned by year. IFREMER is in blue, NCEP in red, and WIS in yellow. The dashed black line shows 0% bias level.

Figure 12 shows model results in a TC-centered frame of reference. On average, within $R = 8$, there is a pattern of underestimation in the left sector and overestimation in the right sector. The notable exceptions are an area of slight overestimation in the directly ahead of the eye (in the left sector) and underestimation near the storm center of all sectors. This patten is robust across two operational hindcasts and two altimeter databases, as shown in Appendix A. This pattern is modulated when binning data in terms of storm attributes. In terms of TC maximum wind speed, the error patterns for both relative bias and nrmse go from weaker to stronger as intensity increases. Moreover, there tends to be an increase in overestimation in the back and left sectors as intensity increases. In terms of TC forward translation velocity, the bias is systematically shifted in a positive direction as translation velocity increases (nrmse changes little). Binning by *rmw* exposed error around the center of TCs; for the smallest storms there was very strong underestimation all sectors accompanied by very high nrmse.

Apparently, wave models perform poorly for small TCs. Certainly, resolution plays a role, as small storms have the strongest spatial gradients. The grids considered here run at 30-arcmin resolutions and thus are affected by errors related to insufficient resolution [56].

Model error also increases with storm intensity. It is known that reanalysis winds tend to understimate TC intensity [8,9]. The issue is compounded because storm size and storm intensity correlated with intensification [57,58] and correlated to one another because TC intensification happens consequently to the contraction of the eye wall (e.g., [59,60]).

Figure 8b shows the coordinates for each LPS, along with their intensity, forward velocity, and direction. Figure 20 shows the joint distribution of wind speed, translation speed, *rmw*, and latitude. These figures show that small, intense storms also tend to translate slowly and occur in lower latitudes (in contrast, the fastest moving storms, V = 15–20 m/s, are only found in higher latitudes). The very strongest storms, Cat 4+, are observed exclusively in the lower latitudes, which explains the unexpected tendency for underestimation in lower latitudes. It is difficult to tease out the effect of *rmw* from intensity, given their overlapping populations. However, their distributions suggest we can describe an archetype that is both likely to occur and prone to model error: a small, intense TC propagating relatively slowly from West to East between 10° and 20° N. The worst case scenario of a fast translating, compact TC [56] is less likely to occur (at least for the TCs sampled in the North Atlantic), and there is some suggestion that an overall increase in relative bias for fast storms might compensate the low bias for strong, compact storms. The systematic increase in bias with storm speed is an unsolved puzzle.

Although the model issues are not directly addressed, it is worth discussing the related literature for clues explaining the poor performance of models in TCs. Already mentioned were resolution and wind fields. Intuitively, systematic underestimation in intensity by reanalysis winds could explain the underestimation of the largest waves in the eye region. However, it is more difficult to make sense of overestimation in the right sector. Might this be explained by deficiencies in wave model physics? There is uncertainty about the nature of air-sea interaction in high wind speeds, particularly momentum flux [61–64]. Adjusting a parametric form of the drag coefficient, $C_D$, in models has improved performance, but wave spectra produced by models in TCs do not provide the right drag for coupled models (e.g., [65,66]). Relevant to our results, Lui et al. 2017 compared source term packages for WW3 and found ST2 underestimated wave heights for $H_s > 6$ m. The underestimation may have been related to the drag coefficient, which was found to max out ($C_{Dmax} = 2.5 \times 10^{-3}$) in winds around 15 m/s. There is some evidence that an entirely new regime of physical exchange emerges (e.g., [26,67], postulated to begin around 40 m/s, mediated by an emulsified ocean interface, where the generation and interaction of spray, spume, foam, bubbles are first-order air–sea interaction processes (e.g., [68,69]). As of now, there is no comprehensive understanding, much less representation of a regime change within the spectral wave model framework, so at worst we extrapolate, and at best, we implement adjusted exchange-coefficients (e.g., [4,66]); the physics, as implemented in source terms, all the while remaining very much the same. TCs also generate copious amounts of rainfall, which present their own effect on the sea surface that we have only recently begun to appreciate ([70,71]), though progress in this area has not yet been implemented in operational models. All operational models use the discrete interaction approximation (DIA) [72] for non-linear interactions. There are numerous documented shortfalls of the DIA, which can be improved by increasing the complexity of the approximation or by calculating the full Boltzmann integral (e.g., [4,73,74]). Another potential effect, neglected by these hindcasts, is wave–current interaction. In coupled model simulations, wind-generated currents are aligned with the wave propagation direction in the right sector, and wave refraction lowers the wave heights in this area (compared to un-coupled models) [75]. This implies that running a coupled model hindcast would likely exacerbate the model bias in the right sector. The Gulf Stream is located in the domain considered here, and it has been shown to focus (or defocus) waves through refraction [76,77], even in coupled model simulations [78].

CBLAST studies showed the tendency of particular wave spectra to appear and dominate the different sectors of a TC: cross-swell in the left sector, following-swell in the right sector, and the opposing-swell in the back sector. Lui et al. (2017) studied Hurricane

Ivan and found consistent overestimation in the left sector (crossing-swell) and in the back sector (opposing-swell). Our Figure 13, which is model-observations pairs within a Cat 1 or higher TC, tells a similar story. This pattern in relative bias was robust, except for the most intense and smallest storms. This suggests that if tuning model physics can address the case of Ivan, similar adjustments may improve storm model results for all but the most intense and smallest storms, where perhaps a new approach is needed. However, prescriptive solutions require further investigation.

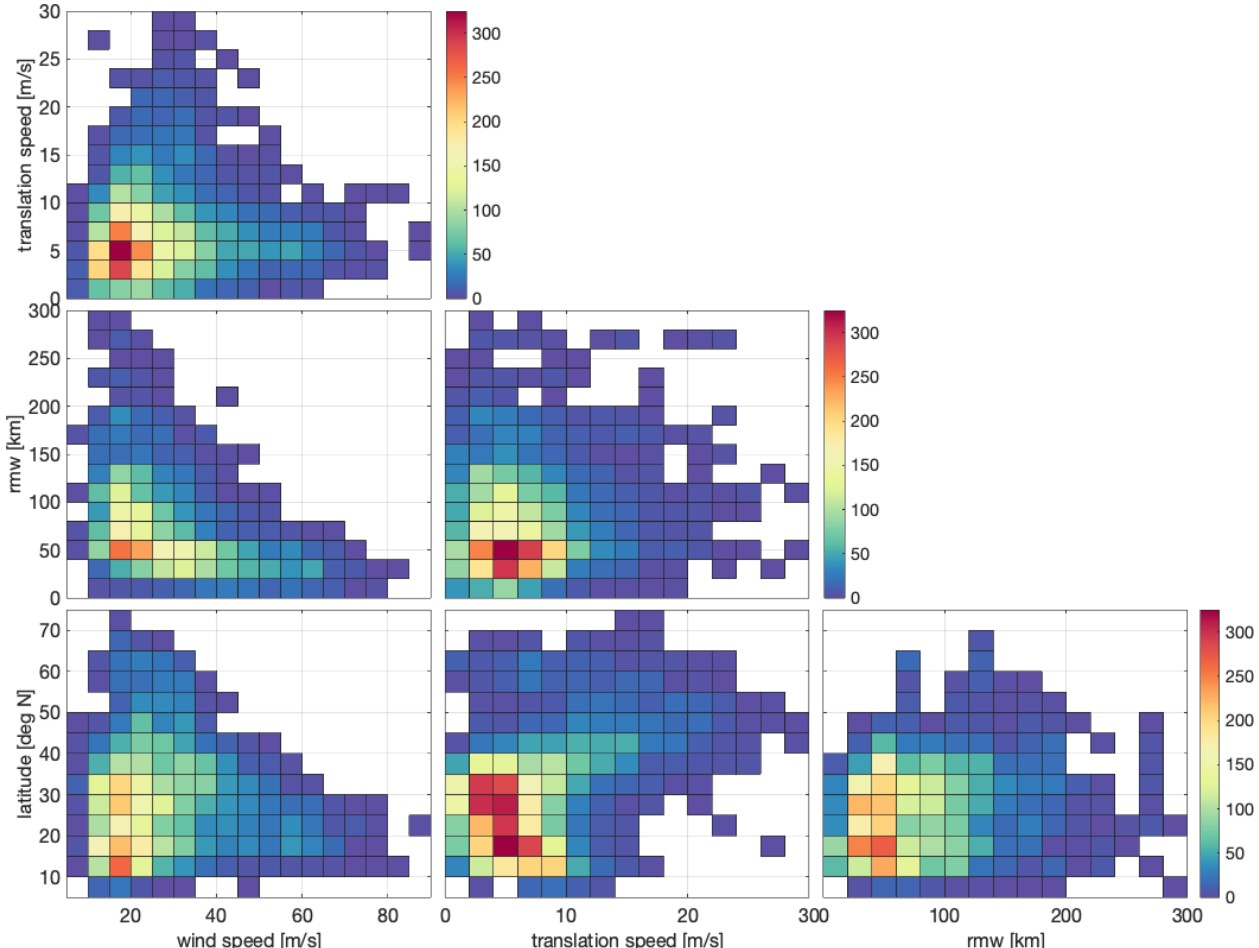

**Figure 20.** Joint distributions of LPS attributes. From left to right is wind speed, translation speed, and radius or maximum winds. From top to bottom is translation speed, radius of maximum winds, and latitude. Note that the top left figure is the same as Figure 2.

We should add that poor performance of an operational hindcasts was not wholly unexpected. While reanalysis winds have been shown to reproduce LPSs, they tend to underestimate intensity compared to IBTRaCS [8]. Backing this up, ref. [79] found CFSR to be deficient in a TC case study [79]. For wave model source terms, the bulk of the data used for tuning and evaluation come from more slowly evolving extra-tropical systems (e.g., [80]). For specific case studies and forecast systems, researchers have gone to great lengths to improve model results in TCs. Almost all studies use specifically developed or otherwise enhanced winds (e.g., [44]) and higher resolution model grids (e.g., [56]). Some studies use fully or partially coupled model systems, where a wave model exchanges information with an atmospheric model or ocean model, or both ([65,75,81–83]. NCEP has even developed a telescoping grid that propagates along with the storm [84]. This list is not meant to be exhaustive, just illustrative of strategies undertaken. Solutions for producing accurate TC generated waves that are also practical in a hindcast framework have yet to be demonstrated.

### 5. Summary

In this study, we report on altimeter observations of significant wave height within TCs, and we evaluate operational hindcasts against these observations. We build on approach of [22], by taking advantage of freely available, curated altimeter data [33]) and hurricane best track information [30–32]. With this approach, we were able to aggregate data to produce composite maps of significant wave heights within a TC-centered frame of reference. Data were then parsed as a function of maximum wind speed and translation velocity. Wave height around a TC is asymmetric reflecting the asymmetry of the wind forcing and the phenomenon of dynamic fetch. Dynamic fetch was observed as wave height was not strictly a function of wind speed but also increased with forward propagation speed, and indeed some of the highest wave heights, on average, were observed within relatively fast translating TCs.

Next, we introduced output from operational wave model hindcasts. NCEP and Ifremer models performed similarly, WIS had higher rmse and lower $R^2$. Statistics are much worse than typical model-altimeter comparisons, and degrade as a function of wind speed and proximity to the storm center. A climatological pattern of error is revealed in the TC centered frame of reference: the underestimation in the left sector and overestimation in the right sector. The notable exceptions were an area of slight overestimation directly ahead of the storm (in the left sector) and near the eye where highest wave heights were consistently underestimated regardless of sector. Our results show that the pattern is robust across most values of storm attributes. It appears to weaken and strengthen with storm intensity and relative bias appears to systematically increase with increasing translational velocity. However, the pattern does not hold for the fastest translating TCs, where we find broad and systematic overestimation of wave heights. Small and intense TCs had expanded area of underestimation, more severe underestimation, and more pronounced random error. The general pattern found here is different to that described in [4], who were able to relate these sectors to particular wave spectral types first suggested in CBLAST [27]. However, their results are consistent with the pattern derived for the most intense storms.

**Author Contributions:** Conceptualization, C.C., P.R., and S.M.; methodology, formal analysis, visualization, and original draft preparation C.C.; review and editing, C.C., P.R., S.M., and T.H.; project administration and funding acquisition, T.H. All authors have read and agreed to the published version of the manuscript.

**Funding:** C.C. and T.H. were funded by the USACE CODS program managed by Spicer Bak.

**Data Availability Statement:** Data used in this study are freely available from the various sources as pointed out in the methods section. Data and code for reproducing plots can be obtained by contacting C.C.

**Acknowledgments:** This study is built upon the hard work of others that collected, quality control, curated, and made available (for free!) satellite, TC, and wave model data. Thanks members of the Scripps/USACE waves group for discussions during the preparation of this work: Alison Ho, Randy Bucciarelli, and James Behrens. Early advice and encouragement came from Chia-Ying Lee, Justin Stopa, Yalin Fan, and Eric Rogers. Jason Sippel provided timely references on TC intensity. Many thanks to two reviewers for helping to improve this work.

**Conflicts of Interest:** The authors declare no conflict of interest.

### Appendix A. Additional Models and Datasets

Although the study focuses on the comparison between the Ifremer global grid and RY19 QC-2 altimeter data, we wanted to show that the pattern of error was robust across models and altimeter data. We tested two different sources of altimeter data and two different quality control levels within one of those datasets: RY19 QC-1 and QC-2 [33] and ESA [34]. For each of these data sources, we evaluated both Ifremer and NCEP hindcasts. Figure A1 shows the relative bias in a TC-centered reference frame in $2R \times 2R$ bins.

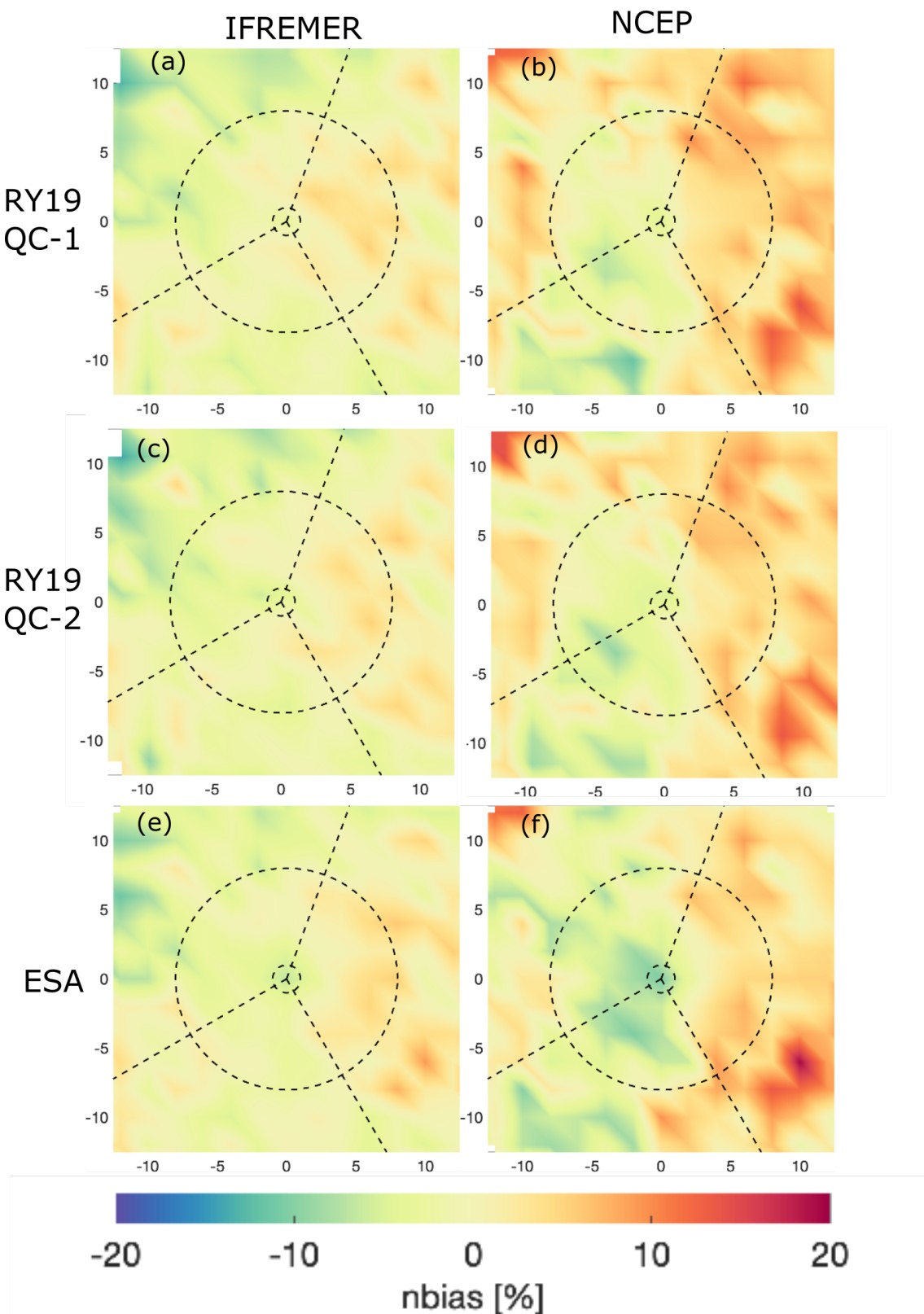

**Figure A1.** Normalized bias in a TC-centered frame of reference. Left column from Ifremer hindcast. Right column from NCEP hindcast. Top to bottom is RY19 QC-1, RY19 QC-2, and ESA altimeter data.

Across data sources, the pattern of error is consistent. There is also good correspondence between the patterns observed in Ifremer and NCEP, although the bias values are higher for NCEP, particularly overestimation in the right sector. Patterns from WIS (not

shown) are consistent across databases, but are not consistent with those of Ifremer and NCEP. Figure A2 shows relative error metrics as a function of altimeter observed wave height. The models fare similarly for for high wave heights. The models tend to underestimate high wave heights, and this underestimation becomes worse as wave heights become higher.

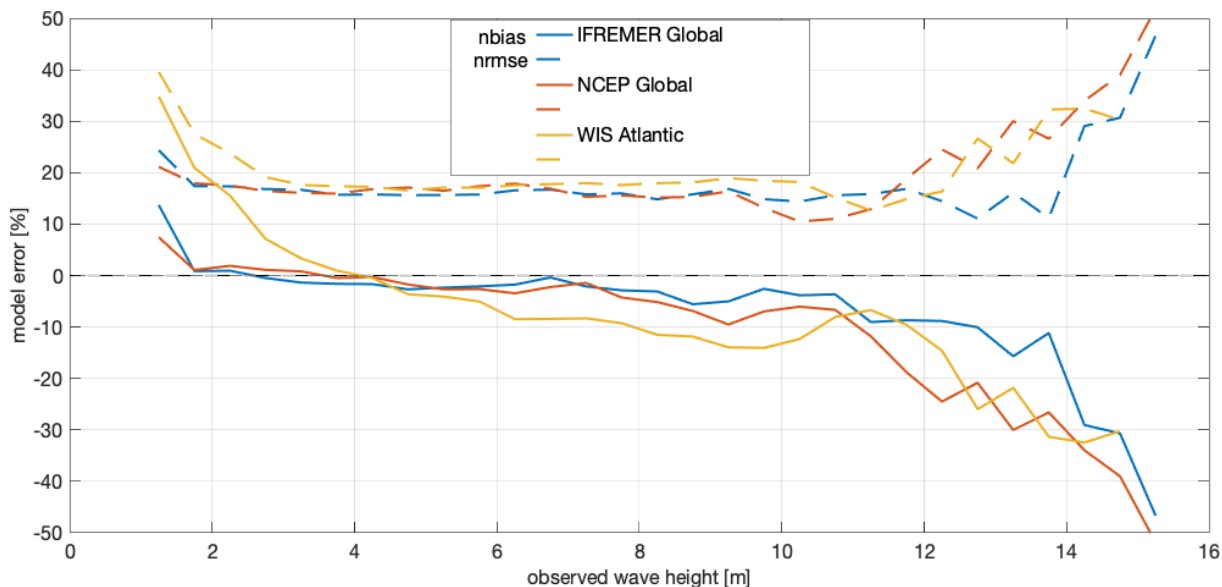

**Figure A2.** Model errors as a function of observed wave height. Relative bias is shown in solid lines and nrmse in dashed lines for Ifremer in blue, NCEP in red, and WIS in yellow.

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
