# Peer review of "Altimeter Observations of Tropical Cyclone-generated Sea States: Spatial Analysis and Operational Hindcast Evaluation"

_jmse, doi:10.3390/jmse9020216_

Round 1
Reviewer 1 Report
It was a real pleasure to read and review your paper

Reviewer 2 Report
Manuscript: Altimeter Observations of Tropical Cyclone Generated Sea States: Spatial Analysis and Operational Hindcast Evaluation
The manuscript presents an analysis of Hs distribution around tropical cyclones in the North Atlantic according to some cyclones characteristics (e.g., intensity, displacement speed, etc) both in satellite data and model. The scientific idea is relevant as pointed out by the authors during the introduction. There is still much information missing about the theme, and this work is an important contribution. However, the manuscript has some relevant issues that must be solved. In this way, I recommend the manuscript for publication, but after major revision.
Main comments:
- After I've read the entire manuscript, I'm sure that the main goal is missing in the introduction. Your introduction promises much more that you delivered, given the idea that the wave generation process within the TC will be further investigated. However, the TC wave field and "extended-fetch" are analysed only using one wave parameter (Hs), which is superficial since the wave spectrum in TC is really complex and change between sectors (maybe you should include Tp and/or wave age). At the end of the day, the focus of the work turned to be the evaluation of the models' performance, which is still a valuable task, but this must be clear in the introduction and in the abstract.
- A lot of data and methods details are missing, which make it difficult to accept some of your affirmation and discussions. I pointed out some of these issues in the specific comments, but, e.g., you say many times that the composites of more intense categories don't have many samples, however, you don't show (1) how is the observation density for each composite, and/or (2) how many TC are sampled in each category. This is important to evaluate if a composite is a representative of not. Maybe you should use a threshold of minimum observations needed to build a composite - is it not clear if you used something like that. As long as you are already using lower quality data in order to have a good overall observation number, it is important to cut off composites that don't have enough samples.
- The discussion is confusing. It is not always clear what the point that you want to expose, and if what are you presenting is based on your results or in someone else results. Sometimes in the discussion, the statements seem to be made entirely based on references. There is not a deep discussion about your results and some statements are made based on the weak analysis. (1) E.g., somewhere in the text, you discuss the relation of the observed bias and error and TC trajectories without even use a reference. You have the tracks, it would be interesting to see the distribution of the tracks of the storms that you analysed. (2) You don't explore the forcing wind field impact on your results. CFSR has some inconsistencies related to changes in the model version, assimilation methods and resolution that potentially impact the wave model results (e.g., the CFSv2 (after 2011) presents improved wind at tropical latitudes). (3) You compare your error metrics with works that do not evaluate model results focusing on extreme weather conditions. This fact must be exposed cleary and the comparison must be made with caution. Many references reported that model performance decrease in extreme conditions, so this impacts a lot the error metrics.
Specific comments:
Abstract: It would be nice that the reader has some previous idea of in which time period the analysis is done in the abstract. It is not clear if you performed some case studies of longer period analysis - and this information changes the impact of the article at a first glance.
Introduction:
line 30: A comma is missing after "e.g." here and all over the text, please review it.
line 43: In the first part of this paragraph a lot of definition and concepts are exposed without a direct reference. It is better to include the references after each main concept (showing from where you got it) rather than put all together at the end of the paragraph. Also, you mention reference 11 only on the footnote and it may be included in the main text.
line 56: "for a nice illustration" sounds informal for a scientific paper. I suggest to avoid it.
line 57: "where rmw is the of maximum winds" --> something is missing, "radius" maybe.
line 72: "they" who? you cited many references before, please specify about each one are you talking about.
line 80: the reference is missing
line 87: Is "This study" referring to Liu et al. (2017) or yours?
Introduction Overview: It is easy to follow. I would suggest a revision of comma and parenthesis usage. However, the goal/scientific question must be relocated in a more objective way. I understood the motivations in lines 28-30, and I suggest that at the end of the introduction you recall this idea to justify your scientific question. In fact, the main question of your work is lost in the text, and should instead be located in the next paragraph, recalling the motivations and concluding your introduction before the last paragraph which summarize the steps of your work.
Methods:
line 105: You are saying that the "forward translation velocity" needed to be calculated in some cases (using sequential position information). How do you calculate it (e.g., with how many time steps)? It is important some consistency in the displacement velocity calculation, considering the given ones and the ones computed by you.
lines 104-108: Here is not clear if your grid is rotated or not (only in results you explain it...) Did you use the mean of instantaneous displacement between time steps? Instantaneous displacement, between only two-time frames, may not be accurate.
lines 115-117: (1) 500 tropical cyclones for the North Atlantic? This information is lost somewhere in the introduction, but you need to define this clearly here in the method section. (2) are you here considering all TCs without any intensity threshold selection? You presented S-S wind scale but you have not explained how you used it.
Table 1: it is important to say that S-S wind scale goes from Category 1 to 5, when the TC is already a Hurricane.
lines 126-127: It is better to include the information in parenthesis in a complete phrase rather than present it in this fragmented way.
lines 141-154: What is the metric of this data quality levels? What is the possible implication of using QC2 data?
line 164: "apples to apples comparison" is too colloquial.
lines 172-173: you need to explain what is WW3 and give references about it and its source terms. (Tolman et al.)
lines 174-187: Datasets references and information are missing here (e.g., Saha et al., Tolman et al.). CFS products changed its resolution and model version in the middle way of your analysis period (2011). It is important to discuss possible impacts caused by this forcing change for both hindcasts, as well to the ST change in WW3.
Results:
line 191: "aka" --> "i.e.,"
lines 194-195: You should put the normalization procedure in the method section.
lines 204-206: which criteria? reference about that?
line 206: how did you calculate this line?
lines 219-223: this should be in the methods section for sure.
line 227: How many samples do you have to perform the composite when you applied this threshold? This is interesting information: how many of your sampled TCs can be classified as hurricanes?
Figure 4: This figure is really nice. I would only suggest increasing the size of the subplots (there are much with space between them). The information about the orange and purple line can be added in the caption.
line 236: it is important to present how many samples each composite has, i.e., how many TC do you have in each category.
line 244: Can the observation of less variation of Hs and V be also a result of fewer samples in these categories? I'm not sure if the composites with low sample size are representative. I think they are important results but the sample size is a key aspect in composites. You could do comment on that in the manuscript.
lines 249-259: This part is confusing and it not clear from where are you getting the information that you are discussing. Is this part of your result or some discussion based on the results of someone else?
Table 3: (caption) "paris" --> pairs (1) what is number od pairs? (2) do you use one value per storm to calculate the errors metric or all value available in a storm?
lines 272: "stats" --> statisics (?)
lines 271-273: Dodet et al. (2020) didn't evaluate the model performance focusing on extreme weather conditions and this must be taken in account when you compare your error metric with them. This is also valid to the discussion between lines 275-279.
line 275-279: Did you performed a statistical analysis see if the differences are "significant" and "significantly"? If not, you should not use these terms here.
lines 280-282: why the sample size alone would increase the error?
Section 3.3.2: here is relevant to discuss the changes in the datasets (e.g., source term, forcing dataset) and how it possibly affected the error over time.
lines 310-312: What do you consider as insufficient sample size? Did you mask in Figure 8 locations where the observation was few? [I believe you didn't by the explanation in line 328, however you would benefit if you do it].
lines 313-317: Since you are not showing the tracks it is nice to include a reference about tropical cyclone density distribution here. Also to the discussion in the lines 325-326.
line 318: There are many observations north from 30ºN. It is possible that you are including sub or extratropical cyclones in your analysis or you separated them?
Discussion:
line 435: It is possible to include how many samples overpass 9 m (maybe include a histogram of the sampled Hs around the cyclones). The composite smooths the field and it is not possible to affirm that the majority of your data is within the altimeters’ functional range only looking to (Fig. 3) (e.g., if you have a broad distribution this may not be so true).
line 446: "err" ?
lines 459-460: this finding is a confirmation of something reported in several studies: the underestimation of Hs higher percentiles. This is reported in climatological analysis and severe weather events, i.e., cyclones. It is good to add here some references (e.g., Stopa and Cheung (2004))
lines 462-463: "physics in the models, should be the same year to year" - but the source term changed
lines 478-480: there is also the impact of the forcing wind field resolution, that sometimes is not enough to reproduce correctly meso and small scale features of the cyclone structure. Some reference about the cyclones representation in reanalysis must be included in the discussion (e.g., Hodges et al., 2017, https://doi.org/10.1175/JCLI-D-16-0557.1)
lines 482-485: I don't remember to see an analysis that related cyclone size, intensity and latitude. How can you affirm that?
lines 486-488: you could select these tracks in your dataset and plot them to prove your point.
lines 511-515: there some new references about how the wave characteristics are affected by the Gulf Stream, e.g., Ponce de León and Guedes Soares (JMSE, 2021).
Round 2
Reviewer 2 Report
The manuscript improved significantly and all of my comments and suggestion were well addressed. I believe the results are more robust and the goal, method limitations and discussion become clearer. There are a few typos that may be corrected during the proof process(e.g., line 34), but not relevant to turn into a new revision round. I am happy to recommend this manuscript to publication, considering that the findings are of great interest to JMSE readers.